# An 800-kyr planktonic $\delta^{18}$O stack for the Western Pacific Warm Pool

Christen L. Bowman[1], Devin S. Rand[1], Lorraine E. Lisiecki[1], and Samantha C. Bova[2]
[1]Department of Earth Science, University of California Santa Barbara, Santa Barbara, CA, 93106, USA
[2]Department of Geological Sciences, San Diego State University, San Diego, CA, 92182, USA
*Correspondence to*: Lorraine E. Lisiecki (lisiecki@geol.ucsb.edu)

**Abstract.** The Western Pacific Warm Pool (WPWP) exhibits different glacial-interglacial climate variability compared to high latitudes, and its sea surface temperatures are thought to respond primarily to changes in greenhouse forcing. To better characterize the orbital-scale climate response covering the WPWP, we constructed a planktonic $\delta^{18}$O stack (average) of 10 previously published WPWP records of the last 800 ka, available at https://doi.org/10.5281/zenodo.10211900 (Bowman et al., 2023), using new Bayesian alignment and stacking software BIGMACS (Lee and Rand et al., 2023). Similarities in stack uncertainty between the WPWP planktonic $\delta^{18}$O stack and benthic $\delta^{18}$O stacks, also constructed using BIGMACS, demonstrate that the software performs similarly well when aligning regional planktonic or benthic $\delta^{18}$O data. Sixty-five radiocarbon dates from the upper portion of five of the WPWP cores suggest that WPWP planktonic $\delta^{18}$O change is nearly synchronous with global benthic $\delta^{18}$O during the last glacial termination. However, the WPWP planktonic $\delta^{18}$O stack exhibits a smaller glacial/interglacial amplitude and less spectral power at all orbital frequencies than benthic $\delta^{18}$O. We assert that the WPWP planktonic $\delta^{18}$O stack provides a useful representation of orbital-scale regional climate response and a valuable regional alignment target, particularly over the 0 to 450 ka portion of the stack.

## 1 Introduction

The tropical Pacific is an important source of heat and moisture to the atmosphere (e.g., De Deckker, 2016; Neale and Slingo, 2003; Mayer et al., 2014) and is thought to have a strong impact on global climate responses during glacial cycles (Lea et al., 2000). Prior studies suggest that the climate of the Western Pacific Warm Pool (WPWP), which is defined by mean annual sea surface temperatures (SST) above 28°C, responds primarily to changes in greenhouse gas concentrations due to the region's large distance from high-latitude ice sheets (Broccoli et al., 2000; Lea, 2004; Tachikawa et al., 2014). Additionally, Earth's orbital cycles cause seasonal variations in insolation or incoming solar radiation, which affect Earth's high and low latitudes differently. In the WPWP, only 0.3°C of SST change is attributed to orbital forcing during the Late Pleistocene (Tachikawa et al., 2014). Thus, climate records of the WPWP region are expected to have features which differ from the high-latitude climate records often used to describe global climate change (e.g., Lisiecki & Raymo, 2005; Past Interglacials Working Group of PAGES, 2016). Here we seek to characterize WPWP climate on orbital timescales and its differences from high-latitude climate, which can help test hypotheses about the sensitivity of the WPWP to orbital forcing, ice volume, and greenhouse gas concentration.

One of the most commonly used paleoceanographic climate proxies is the ratio of oxygen isotopes, denoted as $\delta^{18}$O, in calcium carbonate from foraminiferal tests; this proxy is affected by both water temperature and the $\delta^{18}$O of seawater, which varies with global ice volume as well as local salinity (Wefer and Berger, 1991). The two general types of foraminifera are benthic and planktonic, which live in the deep ocean and surface ocean, respectively. Benthic $\delta^{18}$O is considered a high-latitude climate proxy because deep water temperature is set in high-latitude deep water formation regions and because global ice volume responds primarily to high-latitude northern hemisphere summer insolation. However, planktonic $\delta^{18}$O is influenced by both high-latitude ice volume and local SST and salinity (Rosenthal et al., 2003). Previous studies from the WPWP have shown smaller glacial-interglacial amplitudes of planktonic $\delta^{18}$O change than in benthic $\delta^{18}$O or planktonic $\delta^{18}$O from other regions (Lea et al., 2000; de Garidel-Thoron et al., 2005). This difference has been attributed to smaller sea

surface temperature fluctuations and salinity changes in the WPWP (Broccoli et al., 2000; Lea et al., 2000; de Garidel-Thoron et al., 2005).

Here we present a stack (time-dependent average) of planktonic $\delta^{18}O$ records from ten cores across the WPWP to provide a record of its regional responses over the past 800 ka, which can be compared to the high-latitude response of global and regional benthic $\delta^{18}O$ stacks. The WPWP planktonic $\delta^{18}O$ stack is intended to better characterize orbital responses in WPWP planktonic $\delta^{18}O$ and to improve age models for WPWP sediment cores. Age models for ocean sediment cores, which provide estimates of sediment age as a function of core depth, are commonly constructed by stratigraphic correlation (i.e., alignment) of an individual core's $\delta^{18}O$ record to a global $\delta^{18}O$ stack such as the LR04 or SPECMAP stacks (Linsley and Breymann,
1991; Lea et al, 2000; Chuang et al., 2018; Lisiecki and Raymo, 2005; Imbrie et al., 1984). A stack is the time-dependent average of data from multiple ocean sediment cores that share a common climatic signal, thus increasing the signal-to-noise ratio of the data. Traditionally, stacks have been constructed from global compilations of benthic $\delta^{18}O$ (Lisiecki and Raymo, 2005), planktonic $\delta^{18}O$ (Shakun et al., 2015) or a combination of the two (Imbrie et al., 1984; Huybers and Wunsch, 2004). However, recent studies have advocated the development of regional stacks (Lisiecki and Stern, 2016; Lee and Rand et al.,
2023) to distinguish spatial differences in the timing and amplitude of $\delta^{18}O$ changes.

We constructed a WPWP planktonic $\delta^{18}O$ stack spanning 0 to 800 ka using new Bayesian alignment and stacking software, BIGMACS (Lee and Rand et al., 2023). The new stack consists of previously published planktonic $\delta^{18}O$ data and 65 radiocarbon dates ranging from 1.5 to 36.9 ka from ten cores within the WPWP. We present the new WPWP stack and a
brief comparison of orbital power in the new stack compared to the LR04 global benthic $\delta^{18}O$ stack (Lisiecki and Raymo, 2005) and a recently published stack of regional SST (Jian et al., 2022). We also evaluate the relative timing of WPWP planktonic $\delta^{18}O$ change versus benthic $\delta^{18}O$ change during the last glacial termination.

## 2 Study area

The Western Pacific Warm Pool is a region of the equatorial Pacific with annual average SST between 28℃ to 30℃
(Tachikawa et al., 2014). It covers an area between approximately 15 degrees S to 15 degrees N and 115 to 160 degrees E (Locarnini et al., 2018). Synchronous change in $\delta^{18}O$ is assumed during the stacking procedure (Lee & Rand et al., 2023), so homogeneous conditions in the core locations are important for maintaining the accuracy of the stack. The largely homogeneous WPWP surface ocean makes it a suitable choice for stacking (Lea et al., 2000; Li et al., 2011).

Cores were selected for inclusion based on their location in or near the WPWP, including two cores just beyond the boundary of the typically defined WPWP region. We choose to include these two cores (ODP-1143 and MD05-2930) because of their high resolution and/or an age range that covers the full length of the stack. The cores' locations exhibit oceanographic variability broadly comparable to that observed within the warm pool proper over the period of interest. Core ODP-1143 from the South China Sea, which lies just beyond the northwestern border of the modern WPWP, has an average
annual temperature of ~28℃ and receives northward flowing water from the WPWP during summer (Li et al., 2011). Core MD05-2930 is located along the southern limit of the WPWP in the Gulf of Papua; its SST is primarily controlled by the Australasian Monsoon, with modern SST fluctuating between 26℃ to 29℃ (Regoli et al., 2015). The slightly cooler sea surface temperatures of these two cores are expected to yield slightly more positive $\delta^{18}O$ values than other WPWP sites; these sites may also be sensitive to orbital-scale changes in the WPWP extent.

To evaluate the contribution of temperature to planktonic $\delta^{18}O$ change in the WPWP, we use an Indo-Pacific Warm Pool (IPWP) SST stack (Jian et al., 2022). The IPWP has significant overlap with the WPWP but additionally includes a portion of the Indian Ocean; however, the cores in the Jian et al. (2022) stack are predominantly from the WPWP. The IPWP and WPWP have a mean annual SST of 28 ℃ and 29 ℃, respectively (Locarini et al., 2018).

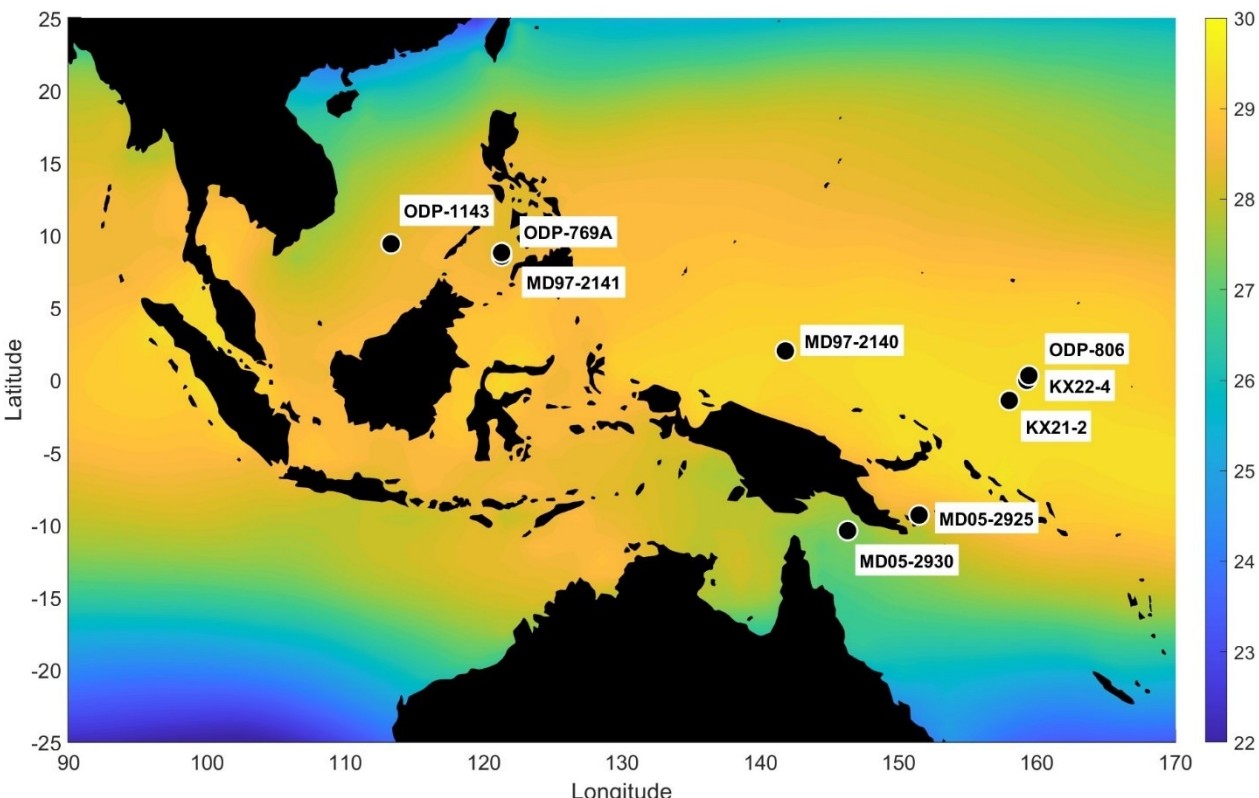


**Figure 1.** Core locations and mean annual sea surface temperatures (°C, color) from 1955-2018 (Locarnini et al., 2018). Created with MATLAB's geoshow() function from the mapping toolbox (The Mathworks Inc., R2023a).

### 3 Data

We compiled previously published planktonic $\delta^{18}O$ measurements from ten tropical Western Pacific cores in or near the
WPWP (**Table 1**). Cores were included in the stack based on their location in the WPWP, an age range spanning at least three glacial cycles, and an average time resolution of at least 4 kyr. Four cores span the last 350 to 500 ka, and six extend back to at least 750 ka (**Fig. 2**). All but one core in the stack uses $\delta^{18}O$ values measured from the planktonic species *Globigerinoides ruber (G. ruber)* sensu stricto (s.s.), whose depth habitat in the WPWP ranges from the upper 45 m to 105 m of the mixed layer depending on how calcification depth is calculated (Hollstein et al., 2017). One core, ODP 1115B, has
data from a different planktonic species, *Trilobatus sacculifer* (formerly *Globigerinoides sacculifer*) whose depth habitat is 20 m to 75 m or potentially as deep as 45 m to 95 m (Sadekov et al., 2009; Hollstein et al., 2017). A species correction of -0.11‰ was applied to the *T. sacculifer* data according to the values presented by Spero et al. (2003). The average mixed layer depth of the WPWP is 50 m to 100 m (Locarini et al., 2018).

A total of 4762 planktonic $\delta^{18}O$ measurements were used to create the stack. The stack spans from 0 to 800 ka; however, there is a significant decrease in data density at 450 ka. Only six cores extend beyond 450 ka, and this portion of the stack is composed of only 960 data points. The lower data resolution results in greater uncertainty and smoothness for the $\delta^{18}O$ features in the older portion of the stack; therefore, we focus our analysis of the stack on the 0 to 450 ka portion.

The stacking algorithm counts each $\delta^{18}O$ measurement equally, so cores with higher resolution have more influence on the stack. The mean sample spacing in the core records used to create the stack ranges from 0.55 to 3.9 kyr, with lower average

sample spacing for the long cores that extend to the older half of the stack. Published data for core MD97-2141 has an average sedimentation rate of 5-15 cm/yr and is sampled at 1 cm intervals with a mean sample spacing of 0.11 kyr (Oppo et al., 2003). However, we smoothed the data using a 5-point running mean sampled every fifth point, which reduces its mean sample spacing to 0.55 kyr, so that this one record does not overly dominate the regional stack. Additionally, we constrain the stack age model using 65 previously published radiocarbon measurements ranging from 1.52 ka to 36.9 ka from five cores (Oppo et al., 2003; Regoli et al., 2015; Lo et al., 2017, Dang et al., 2020, Zhang et al., 2021).

**Table 1.** Locations of cores in the WPWP stack and the temporal coverage of their planktonic $\delta^{18}O$ data.

| Core | Latitude | Longitude | Oldest Age (ka) | Avg Resolution (kyr) | Original Publication |
|---|---|---|---|---|---|
| ODP 1143 | 9.40 | 113.30 | 798.3 | 1.83 | Tian et al., 2006 |
| ODP 769A | 8.79 | 121.29 | 774.1 | 1.22 | Linsley and Breymann, 1991 |
| MD97-2141 [a] | 8.80 | 121.30 | 395.6 | 0.55[b] | Oppo et al., 2003 |
| MD97-2140 | 2.00 | 141.80 | 797.2 | 3.90 | de Garidel-Thoron et al., 2005 |
| MD05-2930 [a] | -10.42 | 146.30 | 799.7 | 2.03 | Regoli et al., 2015 |
| MD05-2925 [a] | -9.34 | 151.46 | 462.7 | 0.76 | Lo et al., 2017; Lo, 2021 |
| ODP 1115B | -9.19 | 151.57 | 798.7 | 2.24 | Chuang et al., 2018 |
| KX21-2 [a] | -1.42 | 157.98 | 394.5 | 0.93 | Dang et al., 2020 |
| KX22-4 [a] | -0.028 | 159.25 | 357.7 | 0.57 | Zhang et al., 2021 |
| ODP 806 | 0.30 | 159.40 | 800.5 | 2.27 | Lea et al., 2000; Medina-Elizalde and Lea, 2005 |

[a] indicates core with radiocarbon data, [b] indicates new resolution after smoothing

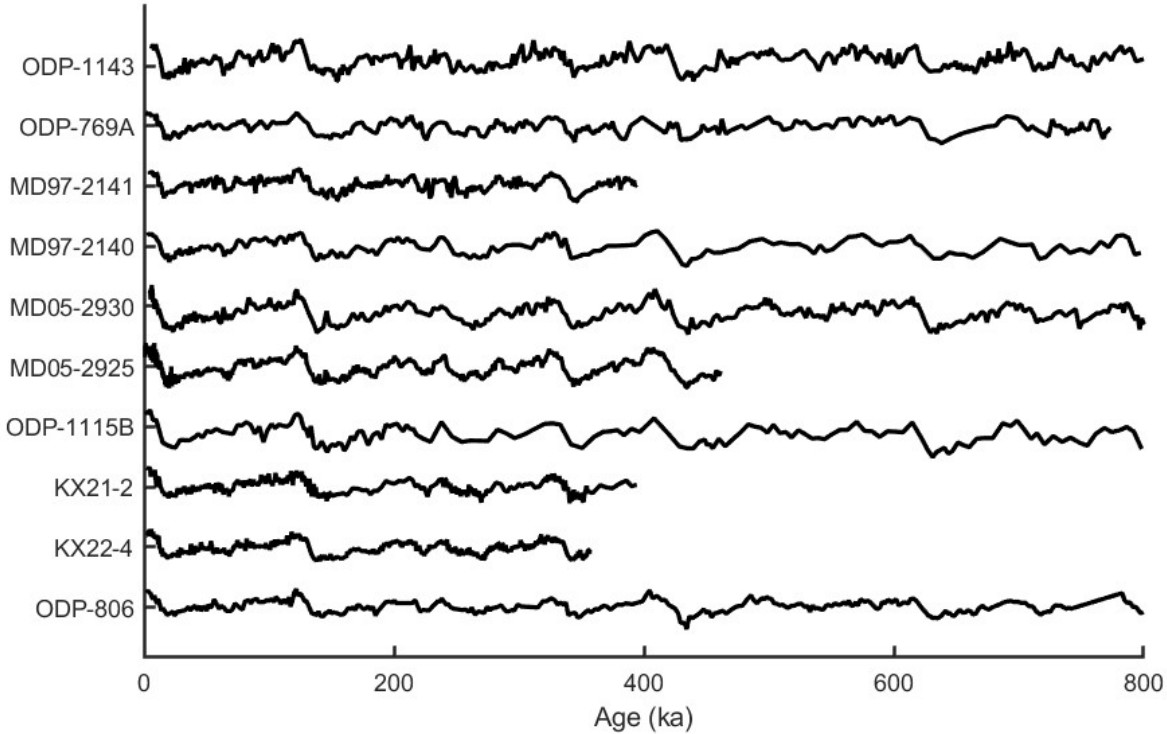

**Figure 2. Planktonic $\delta^{18}$O data of the cores used in the WPWP stack**, plotted on BIGMACS age models for each core and offset vertically. Data are from sites ODP-1143 (Tian et al., 2006), ODP-769A (Linsley and Breymann, 1991), MD97-2141 (Oppo et al., 2003), MD05-2140 (de Garidel-Thoron et al., 2005), MD05-2930 (Regoli et al., 2015), MD05-2925 (Lo et al., 2017; Lo, 2021), ODP-1115B (Chuang et al., 2018), KX21-2 (Dang et al., 2020), KX22-4 (Zhang et al., 2021), ODP-806 (Lea et al., 2000; Medina-Elizalde and Lea, 2005).

# 4 Methods

## 4.1 Stack construction

We use the new Bayesian software package BIGMACS to construct the WPWP planktonic $\delta^{18}$O stack (Lee and Rand et al., 2023). BIGMACS, which stands for Bayesian Inference Gaussian process regression and Multiproxy Alignment for Continuous Stacks, constructs multiproxy age models and stacks by combining age information from both direct age constraints (e.g., radiocarbon data) and probabilistic alignments of $\delta^{18}$O to a target record. Although BIGMACS was developed for benthic $\delta^{18}$O, here we use BIGMACS to align and stack planktonic $\delta^{18}$O; thus, we present analysis to verify the performance of the software for this new application.

BIGMACS stack construction is an iterative process with two steps. In the first step, age models are estimated for each record by aligning to an initial target. Each $\delta^{18}$O record is shifted and scaled to better match the target stack during alignment, and likelihoods assigned to age estimates for each core depth are based on residuals between the core's shifted-and-scaled $\delta^{18}$O value and the target's time-dependent mean and standard deviation. In the second step, a stack is constructed with a Gaussian process regression over all $\delta^{18}$O data using the aligned age models, and the stack's mean and amplitude are set to match the average values of the component records. The new stack is then used as the alignment target to construct age

models for the next iteration, with alignment parameters updated to maximize likelihood using the Expectation Maximization (EM) algorithm. Iterations are performed until convergence. Core-specific shift and scale parameters (**Table 2**) reflect how much each individual record differs from the stack based on the assumption that all records share the same underlying signal but allowing for some scaling or offset based on consistent temperature/salinity gradients within the region as well as foraminiferal species differences (vital effects and depth habitat). Near-homogeneous planktonic $\delta^{18}O$ values between cores (and similar to the final stack) are indicated by shift parameters close to 0 and scale parameters close to 1.

The initial alignment target we used for constructing the WPWP stack was the LR04 stack of 57 globally distributed benthic $\delta^{18}O$ records (Lisiecki and Raymo, 2005) with a constant standard deviation of 0.5 ‰. The original age model for the LR04 stack was created by orbital tuning to a simple ice volume model and has estimated age uncertainties of +/- 4 kyr for the past 800 ka (Lisiecki and Raymo, 2005). Despite the age uncertainty of the LR04 stack and the fact that it reflects benthic rather than planktonic $\delta^{18}O$, it was chosen as the initial alignment target because it is a widely used age model that spans the full 800 kyr time range of the new WPWP stack. Because the stack alignment target is shifted and scaled to match its component records during each iteration, the final stack output by BIGMACS reflects the average WPWP planktonic $\delta^{18}O$ values, rather than the benthic $\delta^{18}O$ values of the initial target.

Importantly, BIGMACS assumes that all records in the stack are homogeneous, i.e., that they all share the same underlying signal (with allowance for site-specific shift and scale values). Under this assumption, all residuals between individual $\delta^{18}O$ measurements and the stack are assumed to reflect variability associated with sampling noise, measurement uncertainty and/or alignment uncertainty. Therefore, when stacking with BIGMACS, it is important to choose records for inclusion in the stack that share the same regional influence. Additionally, because all measurements are treated equally, cores with higher resolution data are more strongly weighted in the stack construction. The stack uncertainty reported by BIGMACS is the time-dependent standard deviation of a Gaussian fit to the $\delta^{18}O$ residuals. To evaluate whether the assumption of homogeneity used by BIGMACS for stack construction is applicable to the WPWP planktonic $\delta^{18}O$ records in our new stack, Section 6.2 compares the WPWP planktonic stack uncertainty and the average alignment uncertainty of the stacked records to results from previously published regional benthic $\delta^{18}O$ stacks.

**4.2 Stack age constraints**

The first 37 kyr of the WPWP are constrained by 65 radiocarbon dates from five cores. We calibrated radiocarbon ages using the Marine20 calibration curve, which uses a model estimate of time-dependent global mean surface reservoir age, with values of ~400 yr in the Holocene and 800 to 1000 yr from 20 to 50 ka (Heaton et al., 2020). We set the reservoir age offset ($\Delta R$) for our sites to 0 yr, meaning we did not change the sites' reservoir ages from the time-dependent Marine20 default. We assigned a 1-sigma uncertainty of 200 yr to the reservoir ages to account for possible changes to the reservoir age offset of the WPWP relative to the Marine20 time-dependent global mean reservoir age. Ages for the remainder of the stack are largely determined by the timing of glacial cycles in the LR04 stack. Thus, the timing of planktonic $\delta^{18}O$ change in the WPWP stack is assumed to be synchronous with the LR04 benthic stack. In section 6.1, we show that there is strong agreement in the timing of planktonic and benthic $\delta^{18}O$ change within WPWP cores and between two benthic stacks from 1.5 to 37 ka, the interval for which WPWP ages are predominantly determined by radiocarbon data.

Core age models were also constrained by age estimates for the first and last $\delta^{18}O$ measurement from each core based on previous publications. Because these previous age estimates were based on a variety of methods, they were assigned a Gaussian uncertainty with a relatively large standard deviation of 4 kyr. Additionally, we added tie points for two cores (ODP-1115B at 75 ka and MD97-2141 at 63 ka and 92.5 ka) to improve the alignment of Marine Isotope Stages (MIS) 3 and 4 to the target stack. Because these tie points were assigned based on identification of stratigraphic features in these two cores compared directly to the target stack, we assigned these age estimates a smaller standard deviation of 1 kyr.

**4.3 Conversions of SST and sea level to isotopic equivalents**

We compare the amplitude of the new WPWP planktonic $\delta^{18}O$ stack with a sea level (ice volume) record and an IPWP SST stack, each of which is converted to the amount of planktonic $\delta^{18}O$ change they are expected to produce. The global sea level stack of Spratt and Lisiecki (2016) was used to calculate an equivalent change in seawater $\delta^{18}O$ due to ice volume ($\Delta\delta^{18}O_{ice}$)

using a conversion of 0.009 ‰ per meter of sea level. This conversion represents the long-term average effect of ice volume change because the size of the effect varies slightly depending on the average $\delta^{18}O$ composition of the ice (Spratt and Lisiecki, 2016).

The IPWP SST stack from Jian et al. (2022) was converted to an $\delta^{18}O$ equivalent using Eq. (1):

$$\Delta\delta^{18}O_{SST} = \frac{-1(SST-29)}{4.8} \tag{1}$$

The 4.8 scaling factor is taken from Bemis et al. (1998). A shift of 29 was chosen to express the effects of SST change relative to a modern WPWP mean SST of 29 °C. Thus, the resulting $\Delta\delta^{18}O_{SST}$ measures change relative to mean annual SST of the WPWP from 1955 - 2018 (Locarini et al., 2018).

### 4.4 Spectral analysis

Power spectral density was calculated to quantify the strengths of response to orbital frequencies in $\delta^{18}O$ for the WPWP and LR04 stacks. Both stacks were sub-sampled at 1 kyr spacing from 0 to 800 ka, and power spectral density was calculated using the multitaper power spectral density estimate function pmtm( ) in MATLAB, with the number of tapers set to two, a rate of one sample per kyr, and an nfft of 512 (The MathWorks Inc., R2023a). Frequencies corresponding to the orbital cycle lengths of eccentricity (100 kyr), obliquity (41 kyr), and precession (23 and 19 kyr) are of particular interest to see how the insolation changes from the cycles affect the $\delta^{18}O$ values.

Normalized power spectral density was also calculated using the same method and sub-sampling for the IPWP SST stack and our WPWP $\delta^{18}O$ stack from 0 to 360 ka to match the age range of the SST stack (Jian et al., 2022). The power spectral density of each record was normalized by dividing by the maximum peak height of the dominant ~100 kyr glacial cycle to evaluate the relative strength of different orbital frequencies.

### 5 Results

The probabilistic stack created by BIGMACS models the planktonic $\delta^{18}O$ value of the WPWP at any point in time as a Gaussian distribution with a time-varying mean and standard deviation. BIGMACS also estimates and applies shift and scale parameters for each core to optimize fit with the stack (**Table 2**). The standard deviation of the stack reflects scatter in the shifted-and-scaled $\delta^{18}O$ measurements at each point in time, including the effects of statistical uncertainty in each core's age model. The standard deviation of the stack does not include any information about absolute age uncertainty (outside the range of radiocarbon). However, uncertainty does increase where data are sparse. The standard deviation of the new WPWP planktonic $\delta^{18}O$ stack has an average value of 0.19 ‰ for the full stack and 0.17 ‰ for 0 to 450 ka, where data are more densely spaced.

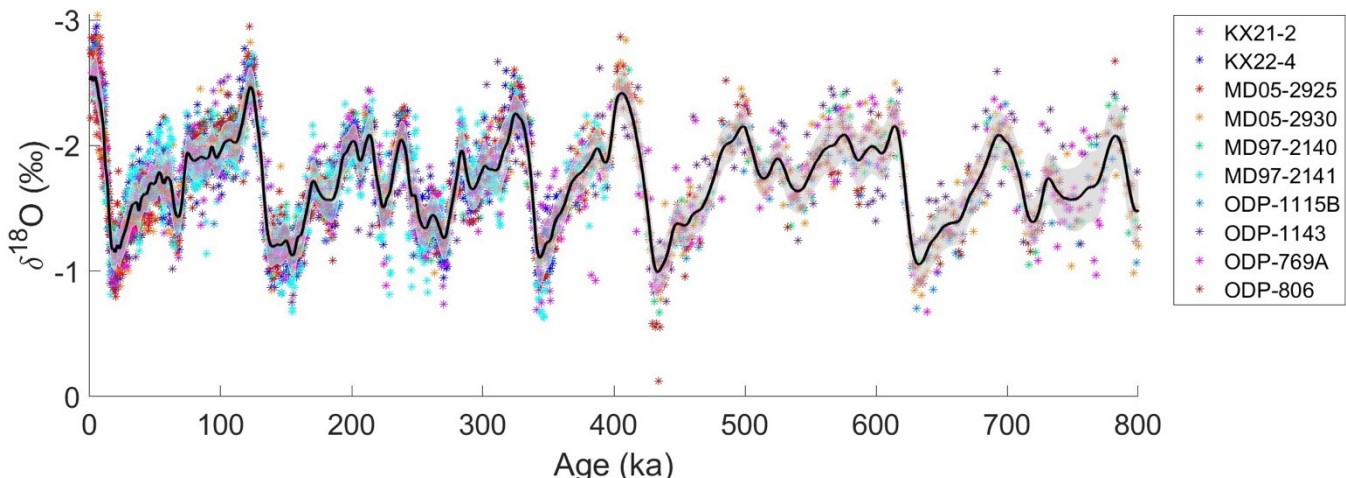

205

**Figure 3. The WPWP planktonic $\delta^{18}$O stack mean (black) and 1 standard deviation (gray shading).** Colored asterisks show the planktonic $\delta^{18}$O measurements from each core after applying the core-specific shift and scale parameters calculated during alignment. Data from MD97-2141 was smoothed and sampled at one-fifth the originally published resolution.

210 **Table 2**. Core-specific shift and scale parameters and the standard deviation of $\delta^{18}$O residuals between the WPWP stack and each core (after applying the estimated shift and scale parameters).

| Core | Shift | Scale | Residual $\sigma$ (‰) |
|------|-------|-------|----------------------|
| **ODP 1143** | -0.38 | 1.09 | 0.23 |
| **ODP 769A** | -0.53 | 0.81 | 0.72 |
| **MD97-2141** | -0.62 | 0.8 | 0.20 |
| **MD97-2140** | -0.23 | 0.92 | 0.17 |
| **MD05-2930** | 0.46 | 1.1 | 0.18 |
| **MD05-2925** | 0.23 | 1.11 | 0.15 |
| **ODP 1115B** | 0.64 | 1.14 | 0.17 |
| **KX21-2** | -0.24 | 0.83 | 0.18 |
| **KX22-4** | -0.19 | 0.79 | 0.16 |
| **ODP 806** | -0.35 | 0.72 | 0.18 |

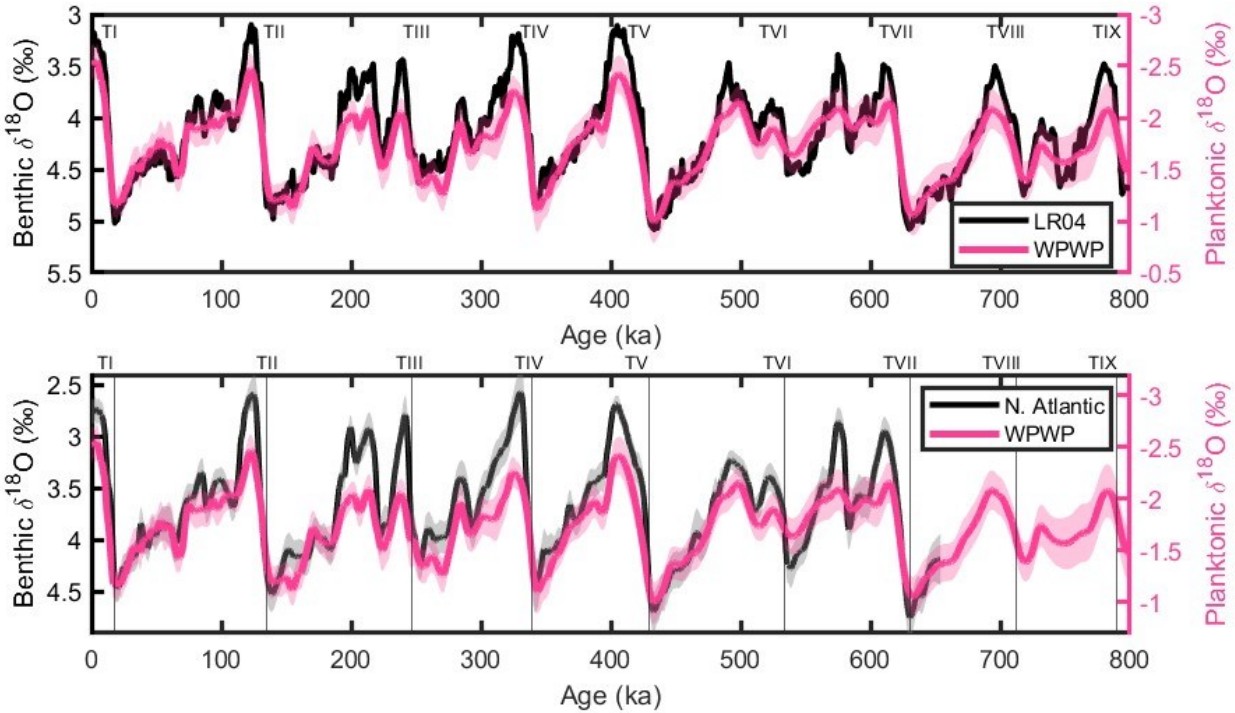

**Figure 4. Stack comparisons.** (top) Comparison between our WPWP planktonic $\delta^{18}$O stack (pink) and the global LR04 benthic $\delta^{18}$O stack (black) (Lisiecki and Raymo, 2005). (bottom) Comparison between our WPWP planktonic $\delta^{18}$O stack (pink) and a regional North Atlantic benthic $\delta^{18}$O stack (black) (Hobart et al., 2023). Shaded error bars represent one standard deviation in the WPWP and North Atlantic stacks. Glacial terminations are labelled with vertical lines based on ages for TI - TVII from Hobart et al. (2023); and TVIII-TIX from Lisiecki and Raymo (2005).

The WPWP planktonic stack has weaker glacial-interglacial amplitudes than the global LR04 benthic $\delta^{18}$O stack (Lisiecki and Raymo, 2005) or a North Atlantic benthic $\delta^{18}$O stack produced by BIGMACS (Hobart et al., 2023). The average glacial-interglacial amplitude for Terminations I to V is 1.7 ± 0.1 ‰ and 1.8 ± 0.1 ‰ in the LR04 and North Atlantic benthic stacks, respectively, but only 1.2 ± 0.1 ‰ in the WPWP planktonic stack. (The reported one standard deviation uncertainty for the mean amplitude of each stack is calculated using the time-dependent standard deviation of $\delta^{18}$O in each stack.) This amplitude difference is also reflected in the spectral analysis of the stacks. Across all three orbital frequencies, there is greater spectral power in the LR04 benthic stack than the WPWP planktonic stack (**Fig. 5**).

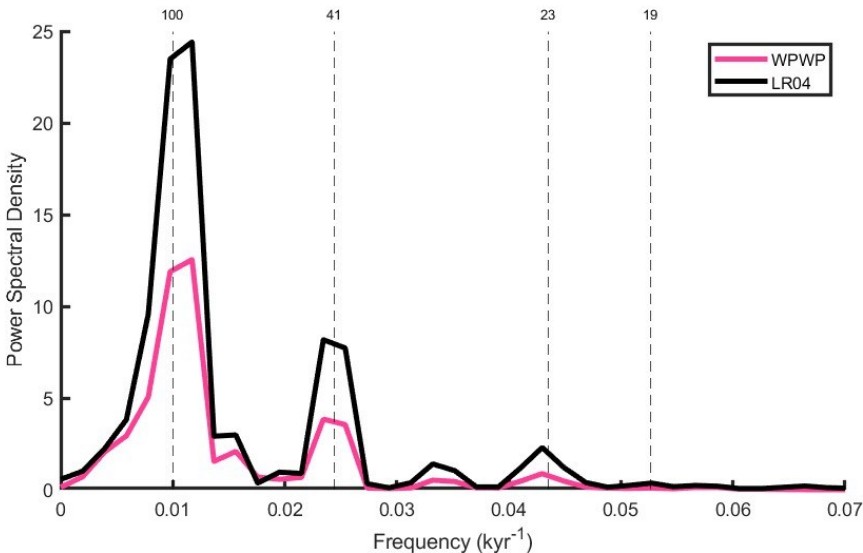

**Figure 5. Power spectral density of the WPWP planktonic $\delta^{18}$O stack (pink) and the LR04 benthic $\delta^{18}$O stack (black).** Spectral power is calculated from 0 to 800 ka for both stacks using MATLAB's pmtm( ) function (The Mathworks Inc., R2023a). Orbital frequencies that correspond to 100, 41, 23 and 19 kyr are labeled with vertical dashed lines.

# 6 Method validation and limitations

## 6.1 Age model assumptions

The use of the LR04 stack as an initial alignment target for our WPWP stack assumes benthic and planktonic $\delta^{18}$O are changing synchronously; however, the signals recorded by benthic and planktonic $\delta^{18}$O could differ due to either a different transit time of the global ice volume signal to the deep ocean compared to surface of the WPWP and/or due to asynchronous temperature and salinity changes between the WPWP and high-latitude deep water formation regions. To evaluate potential timing differences in the two signals, we compare the age model for the portion of the WPWP planktonic stack constrained by radiocarbon data (1.5 to 37 ka) to the equivalent portion of the LR04 and LS16 global benthic $\delta^{18}$O stacks (Lisiecki and Raymo, 2005; Lisiecki and Stern, 2016). The LS16 global stack is constructed with direct $^{14}$C age constraints and is weighted towards the Pacific based on ocean basin volume, whereas the LR04 stack is based only on indirect age constraints and more heavily weighted toward Atlantic values (Lisiecki and Raymo, 2005; Lisiecki and Stern, 2016). However, all three stacks show good agreement for the timing of $\delta^{18}$O change during Termination I, suggesting that age estimates for WPWP planktonic and benthic $\delta^{18}$O are similar on orbital time scales (**Fig. 6**). We also compare changes in planktonic and benthic $\delta^{18}$O measured within individual WPWP cores as a function of depth for MD05-2925, ODP-1143, and ODP-806 (Lo et al., 2019; Lo, 2021; Tian et al., 2006; Lea et al., 2000; Medina-Elizalde and Lea, 2005; Bickert et al., 1993). These cores do not show a consistent lead/lag between the planktonic $\delta^{18}$O and benthic $\delta^{18}$O records (**Fig. S1 - S3**), additionally indicating that the timing of WPWP planktonic and benthic $\delta^{18}$O change is similar on orbital time scales.

The relative timing of millennial-scale variability between the WPWP planktonic stack and benthic $\delta^{18}$O is more difficult to evaluate. Apparent differences in timing of a millennial-scale feature in the stacks between 36 and 38 ka may be an artifact of age model uncertainty. Age uncertainty for the LR04 stack beyond 30 ka is ± 4 kyr, and age estimates for the LS16 stack have a 95% confidence interval width of 2 to 4 kyr between 30 and 40 ka. Age estimates for that portion of our WPWP stack are not well constrained due to the scarcity of radiocarbon data available beyond 30 ka. The portions of our WPWP stack older than 37 ka, which are not constrained by radiocarbon data, inherit the +/- 4 kyr age uncertainty of the LR04 stack used

as the initial alignment target. Thus, we have no independent age estimates for WPWP planktonic $\delta^{18}$O change older than 37 ka.

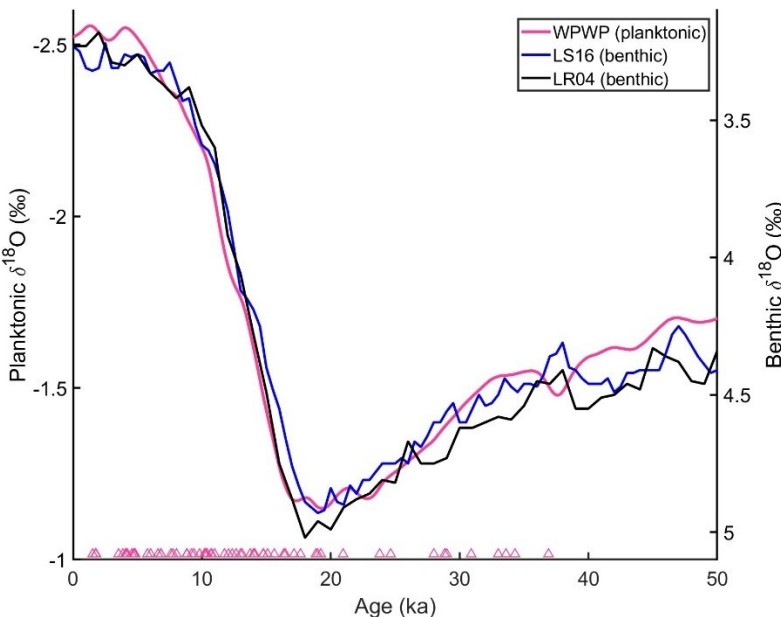

**Figure 6. The WPWP planktonic $\delta^{18}$O stack (pink) compared to the global benthic $\delta^{18}$O stacks** of Lisiecki and Stern (2016) (blue) and Lisiecki and Raymo (2005) (black). Triangles represent radiocarbon ages included in our WPWP stack construction (Oppo et al., 2003; Lo et al., 2017; Regoli et al., 2015; Dang et al., 2020; Zhang et al., 2021).

## 6.2 Application of BIGMACS to planktonic $\delta^{18}$O

### 6.2.1 Standard deviation of planktonic versus benthic stacks

The new Bayesian alignment software BIGMACS has previously only been applied to benthic $\delta^{18}$O data (Lee and Rand et al., 2023), and this study is the first to use the software to stack planktonic $\delta^{18}$O data. To evaluate the performance of BIGMACS in stacking WPWP planktonic $\delta^{18}$O, we compare the average standard deviation of the new WPWP planktonic $\delta^{18}$O stack to two benthic $\delta^{18}$O stacks constructed with BIGMACS using six cores from the Deep Northeast Atlantic (DNEA stack) and four cores from the Intermediate Tropical Western Atlantic (ITWA stack). Based on the BIGMACS assumption of homogeneity across aligned records, all $\delta^{18}$O residuals are assumed to be internal errors associated with sampling noise and measurement uncertainty and, thus, all residuals contribute similarly to estimating the stack's time-dependent standard deviation. A similar standard deviation for $\delta^{18}$O in the stacks would indicate a similar signal-to-noise ratio in the stacked data, suggesting a similar effectiveness in the stacking process.

The DNEA and ITWA stacks have mean standard deviations of 0.13 ‰ and 0.2 ‰, respectively, for 0 to 60 ka (Lee and Rand et al., 2023), while the new WPWP planktonic stack has a mean standard deviation of 0.16 ‰ for the same age range. A larger mean standard deviation of 0.19 ‰ for the full age range of 0 to 800 ka for our WPWP stack is likely due in part to the lower resolution of data used in the second half of the stack; however, it is still similar to the standard deviation of the ITWA benthic stack. The similar $\delta^{18}$O standard deviations for the planktonic and benthic stacks suggests that BIGMACS may be similarly effective at aligning and stacking homogeneous regional planktonic $\delta^{18}$O data as regional benthic $\delta^{18}$O data. However, before stacking either benthic or planktonic $\delta^{18}$O records, BIGMACS users should carefully evaluate whether the records to be aligned and stacked are homogeneous (i.e., share a common and synchronous signal).

### 6.2.2 Homogeneity of WPWP planktonic $\delta^{18}O$

The process of stack construction assumes that all included WPWP planktonic $\delta^{18}O$ records have one homogeneous signal, i.e., that they all share the same underlying signal (with allowance for site-specific shift and scale values caused by physical processes, such as temperature and salinity gradients between the core locations). The shift and scale values calculated for each record during stack construction can be used as an estimate of how similar or different the means and amplitudes of the planktonic $\delta^{18}O$ signals are between cores. The shift values of the 10 cores in the WPWP stack range between -0.62 and 0.64 ‰, and scale values range from 0.72 to 1.11 (**Table 2**). The benthic DNEA and ITWA stacks constructed using BIGMACS have a smaller range of shift and scale parameters than the WPWP stack. Shift values range from -0.07 to 0.25 ‰ and -0.25 to 0.3 ‰ for the DNEA and ITWA stacks, respectively; benthic scale values range between 0.92 to 1.04 for the DNEA stack and 0.91 to 1 for the ITWA stack (Lee and Rand et al., 2023). Thus, core-specific shift and scale values suggest more spatial variability in WPWP planktonic $\delta^{18}O$ than in regional benthic $\delta^{18}O$ compilations.

WPWP cores ODP-769A and MD97-2141, both of which are located in the Sulu Sea, have two of the largest negative shifts and smallest scale values (Linsley et al., 1991; Oppo et al., 2003). Cores KX21-2, KX22-4 and ODP-806, located in the eastern, open ocean portion of the WPWP, also have small scale values and negative shifts (Dang et al., 2020; Zhang et al., 2021; Lea et al., 2000; Medina-Elizalde and Lea., 2005). The similar shift and scale values for neighboring cores with different data resolution suggest that these results reflect real differences in SST or salinity variability within the WPWP and indicate a weaker amplitude for planktonic $\delta^{18}O$ change at these sites. Previous studies show regional differences in $\delta^{18}O_{seawater}$ that may explain the reduced amplitude of planktonic $\delta^{18}O$ change at sites in the Sulu Sea and eastern WPWP (Lea et al., 2000; de Garidel-Thoron et al., 2005). Unlike the central and southern WPWP where glacial surface water $\delta^{18}O$ shifted toward more positive values at the LGM (Visser et al., 2003; Xu et al., 2008, Li et al. 2016), sites ODP-769A, MD97-2141, KX21-2, KX22-4 and ODP-806 show negative shifts in surface water $\delta^{18}O$ at the LGM (Rosenthal et al., 2003; Lea et al., 2000). The observed heterogeneity in $\delta^{18}O_{seawater}$ likely results from regional differences in precipitation (de Garidel-Thoron et al., 2005) and/or the varied impacts of changes in sea level on the Indonesian throughflow and connectivity of regional seas (Linsley et al., 2010).

Although alignment of $\delta^{18}O$ signals for stacking requires an assumption that the WPWP planktonic $\delta^{18}O$ is homogenous, the BIGMACS estimated core-specific shift and scale parameters should still allow us to extract the underlying signal common to the region despite small differences in the mean and amplitude of the signal among core sites. The similar standard deviation for the planktonic stack compared to benthic stacks suggests that the shift and scale factors are effective for identifying a common, shared planktonic $\delta^{18}O$ signal across the WPWP.

More variability in the planktonic $\delta^{18}O$ data is expected because the surface ocean composition has greater spatial variability due to factors like temperature and salinity than the deep ocean, which could account for some of the disparity in shift and scale values of the benthic versus planktonic data. The greater spatial variability in planktonic data is one reason why regional planktonic stacks are more useful than global planktonic stacks. By describing regional patterns of response, regional planktonic stacks can improve age models based on stratigraphic alignment. The higher resolution, 0 to 450 ka portion of our WPWP stack may be particularly useful for this purpose. Although the new WPWP planktonic stack can improve estimates of relative age regionally, we caution that its absolute ages are susceptible to our assumption of synchronous change in benthic $\delta^{18}O$ and WPWP planktonic $\delta^{18}O$ and the absolute age uncertainty of the LR04 stack.

### 6.2.3 Planktonic vs. benthic alignment uncertainty

Here we compare the age uncertainty during planktonic versus benthic alignment in BIGMACS. The average 95% confidence interval width for alignment uncertainty across all WPWP cores is 4.8 kyr for the full length of the WPWP stack and 4.4 kyr for the 0 to 450 kyr portion of the stack, which has higher resolution data. Similarly, a North Atlantic benthic $\delta^{18}O$ stack also constructed using BIGMACS has an average alignment uncertainty of 4.4 kyr for the 0 to 654 ka length of that stack (Hobart et al., 2023). Thus, despite differences in the amplitude and spatial variability of WPWP planktonic $\delta^{18}O$ compared to North Atlantic benthic $\delta^{18}O$, alignment uncertainty is similar for the construction of planktonic and benthic

330    $\delta^{18}O$ stacks. During most of the stack construction process, all records used for alignment are exclusively planktonic or benthic $\delta^{18}O$. (Although the LR04 benthic stack is used as the initial alignment target for the WPWP stack, the BIGMACS alignment target is updated to reflect the mean planktonic $\delta^{18}O$ signal during each iteration of stack construction.)

335    BIGMACS assumes that the records used for alignment share the same underlying signals; therefore, alignment should be more reliable with smaller uncertainties when a nearby planktonic $\delta^{18}O$ record is aligned to the WPWP planktonic stack rather than the LR04 benthic stack. We demonstrate the potential impacts of aligning to different stacks by comparing the age estimates for the planktonic $\delta^{18}O$ record of core MD01-2378 (Holbourn et al., 2005) from the Timor Sea (slightly outside the boundaries of the WPWP) based on alignment to either the WPWP stack or the LR04 stack (Bowman et al., 2023). Differences between the features of the two stacks during MIS 3 and 4 produce a ~14 kyr error in the alignment of the

340    core to the LR04 stack, as indicated by the shifted position of the dashed vertical line in **Fig. 7**. The proper alignment of MIS 4 to the WPWP stack produces a 95% CI width of 6.5 kyr for estimated age at that time, compared to a 95% CI width of 13 to 18 kyr associated with the incorrect alignment to the LR04 stack. Because the planktonic $\delta^{18}O$ records near the WPWP share features which differ from those of benthic $\delta^{18}O$, age model results for WPWP cores should be more accurate when their planktonic $\delta^{18}O$ records are aligned to the WPWP stack than to a benthic stack.

345

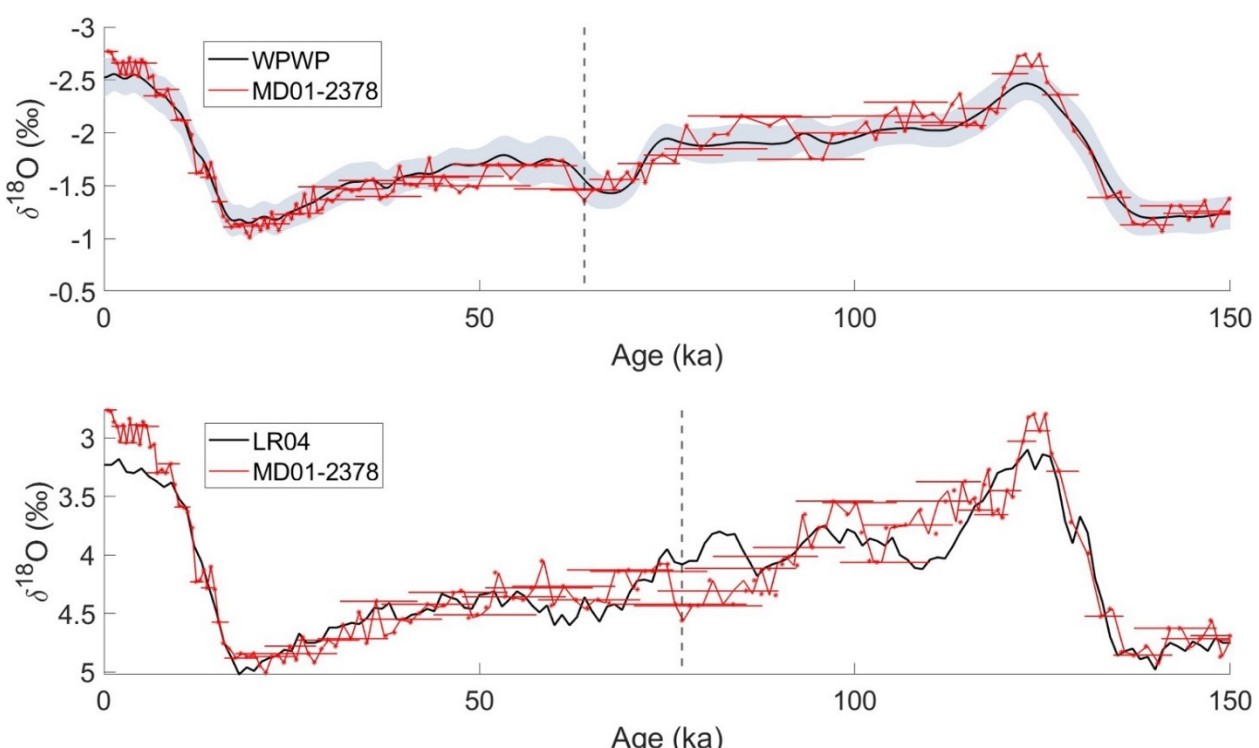

**Figure 7. MD01-2378 alignment target comparison.** (top) Planktonic $\delta^{18}O$ for core MD01-2378 (red, Holbourn et al., 2005) aligned to the WPWP planktonic $\delta^{18}O$ stack (black, with gray shading for 1-standard deviation in $\delta^{18}O$). (bottom) Planktonic $\delta^{18}O$ for MD01-2378 (red, Holbourn et al., 2005) aligned to the benthic $\delta^{18}O$ LR04 stack (black, Lisiecki and Raymo, 2005). In both panels, red symbols mark

350    MD01-2378 planktonic $\delta^{18}O$ samples with shift and scale applied to match the respective alignment targets. Horizontal error bars indicate the 95% CI alignment uncertainty for every third $\delta^{18}O$ measurement (to improve figure legibility). The vertical dashed lines mark two different alignments of 8.81 m depth in MD01-2378, which shifts from 64 ka when aligned to the WPWP planktonic stack to 77 ka when aligned to the LR04 benthic stack.

## 6.3 Contributions of SST and ice volume to WPWP planktonic $\delta^{18}$O

A recent study by Jian et al. (2022) constructed an IPWP SST stack from 0 to 360 ka. We compare the orbital-scale variability between the SST stack and our planktonic $\delta^{18}$O stack using spectral analysis and by converting the SST stack change to $\Delta\delta^{18}O_{SST}$, which is an estimate of the oxygen isotope fractionation in foraminiferal carbonate caused by SST. Many of the same features can be seen in the WPWP stacks of planktonic $\delta^{18}$O and SST (**Fig. 8, top**); however, the planktonic $\delta^{18}$O values of the last two interglacials are similar to one another whereas Holocene SST is notably cooler than the SST of the penultimate interglacial. The WPWP planktonic $\delta^{18}$O and SST stacks have similar proportions of normalized spectral power at orbital frequencies, but SST has slightly less obliquity power and slightly more precession power (**Fig. 9**).

To estimate the combined effects of SST and ice volume change, the IPWP $\Delta\delta^{18}O_{SST}$ was added to an estimate of $\Delta\delta^{18}O_{ice}$ (**Fig. 8, bottom**) from a global sea level (ice volume) stack (Spratt and Lisiecki, 2016). The combined SST and ice volume $\Delta\delta^{18}O_{SST+ice}$ should represent the majority of the planktonic $\delta^{18}$O change in our WPWP planktonic $\delta^{18}$O stack, except for salinity-induced changes. The two combined $\Delta\delta^{18}$O components show similar glacial/interglacial cyclicity and timing of change as our WPWP planktonic $\delta^{18}$O stack but with a slightly larger amplitude, particularly during MIS 2 and 3 (20 to 70 ka), MIS 7 (200 to 250 ka) and MIS 9 (310 to 330 ka). Salinity-induced changes in the $\delta^{18}$O of WPWP surface water may offset some of the WPWP $\Delta\delta^{18}O_{SST+ice}$ signal. However, some of the discrepancy could also be explained by spatial variability in the IPWP if the sites used for our WPWP planktonic $\delta^{18}$O stack had less average SST change than those in the IPWP SST stack (Jian et al., 2022).

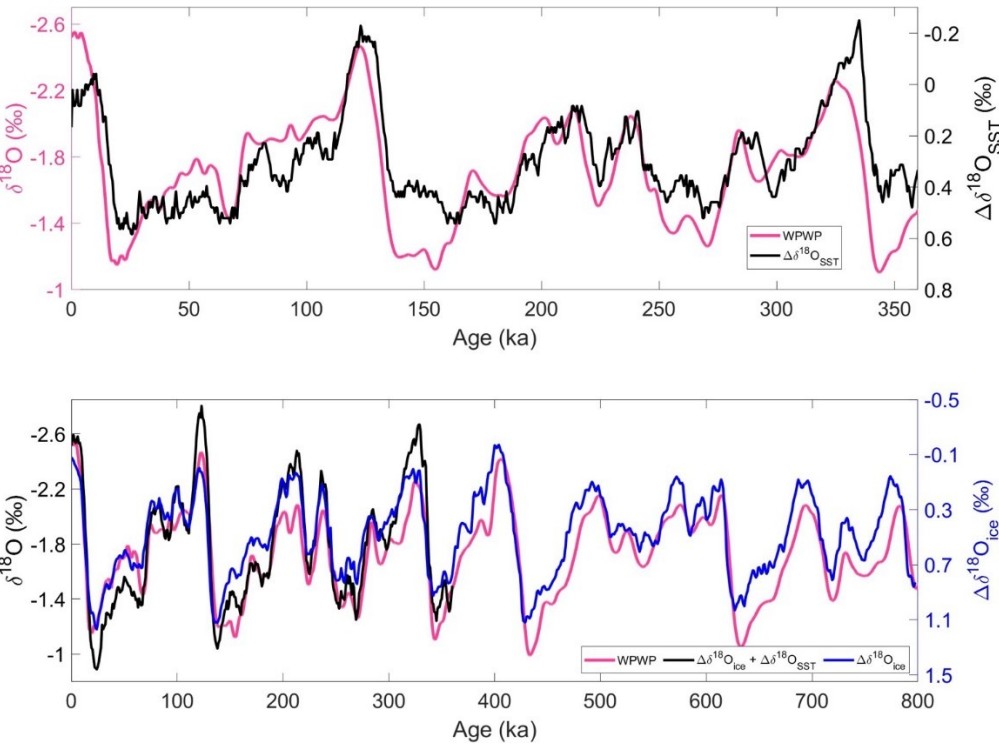

**Figure 8. Ice volume and temperature contributions to WPWP planktonic $\delta^{18}$O.** (top) The WPWP planktonic $\delta^{18}$O stack (pink) and the IPWP SST stack of Jian et al. 2022, on its original age model and converted to $\Delta\,\delta^{18}$O per mil-equivalent (black). (bottom) The WPWP stack (pink) compared to a sea level stack (Spratt and Lisiecki, 2016) converted to $\Delta\delta^{18}O_{ice}$ per mil-equivalent (blue), and the sum of $\Delta\delta^{18}O_{SST}$ and $\Delta\delta^{18}O_{ice}$ change (black).

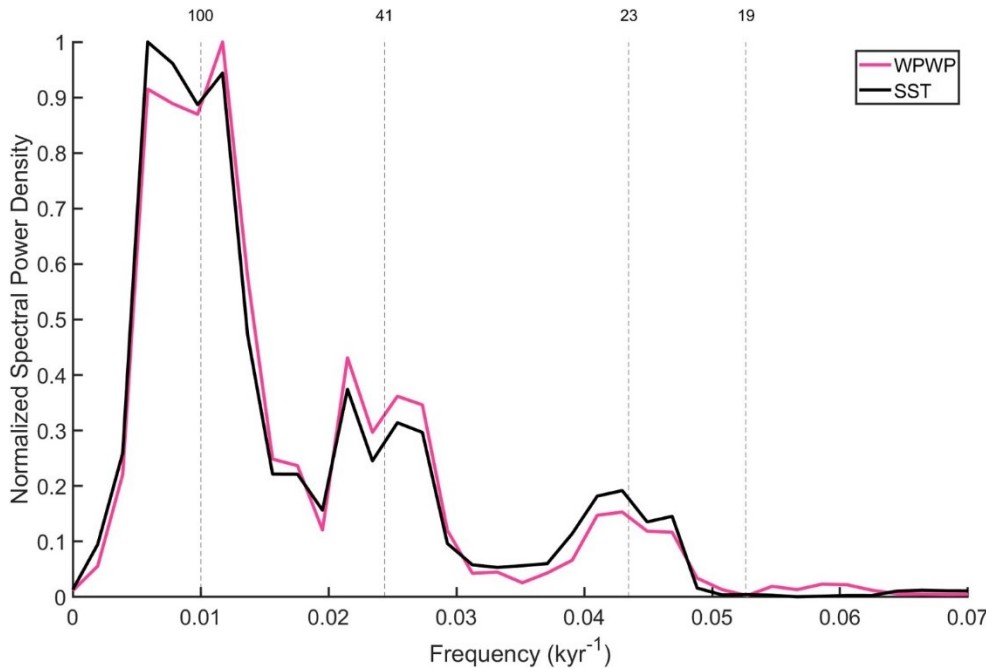

**Figure 9. Normalized power spectral density** of the WPWP planktonic $\delta^{18}$O stack (pink) and IPWP SST stack (black, Jian et al., 2022) from 0 to 360 ka for both stacks using MATLAB's pmtm( ) function (The Mathworks Inc., R2023a). Orbital frequencies that correspond to 100, 41, 23 and 19 kyr are marked by vertical dashed lines.

## 7 Data availability

The WPWP planktonic $\delta^{18}$O stack be accessed on Zenodo with the DOI https://doi.org/10.5281/zenodo.10211900 (Bowman et al., 2023) in the "WPWP_planktonic_stack.txt" file, which contains the age, mean $\delta^{18}$O, and the $\delta^{18}$O standard deviation. The same file is also available as "stack.txt" in the WPWP_10cores_v12_stack_output.zip file. The previously published depth and planktonic $\delta^{18}$O data as well as any radiocarbon or tie points for each core used during stack construction can be found as .txt files in the 'Inputs' folder. BIGMACS-produced age models, depth, calibrated radiocarbon ages, and planktonic $\delta^{18}$O data for each core can be found in the 'Outputs' folder in the "results.mat" file and as .txt files in the individual folders named for each core.

The planktonic $\delta^{18}$O alignments for core MD01-2378 (Holbourn et al., 2005) to both the WPWP planktonic $\delta^{18}$O stack and the LR04 benthic $\delta^{18}$O stack (Lisiecki and Raymo, 2005) can also be accessed at the DOI https://doi.org/10.5281/zenodo.10211900. The BIGMACS-produced age model and the data used for the alignment can be found in the "results.mat" file of the "MD01-2378_WPWPalignment_output.zip" (or "…LR04alignment_output") files. The age model results can also be found in text file form in the "Ages" folder of the output folders. The depth, radiocarbon, and planktonic $\delta^{18}$O data used for alignments can be found in text file form in the "MD01-2378" folder of the "MD01-2378_WPWPalignment_input.zip" (or "…LR04alignment_input") files.

The alignment software BIGMACS (Lee and Rand et al., 2023) used to construct our WPWP stack can be downloaded at https://github.com/eilion/BIGMACS.

## 8 Conclusions

We present a regional planktonic $\delta^{18}O$ stack of the Western Pacific Warm Pool constructed from ten previously published cores using the new alignment software BIGMACS (Lee and Rand et al., 2023). The stack age model is constrained by 65 radiocarbon dates from 1.5 to 37 ka in four WPWP cores and otherwise follows the age model of the LR04 benthic $\delta^{18}O$ stack (Lisiecki and Raymo, 2005). Within the radiocarbon time interval, the timing of WPWP planktonic $\delta^{18}O$ appears nearly synchronous with global mean benthic $\delta^{18}O$ change. The WPWP planktonic $\delta^{18}O$ stack provides a useful regional alignment target for WPWP planktonic $\delta^{18}O$ records, particularly for the 0 to 450 ka portion which has a higher resolution than the older portion of the stack. Future improvements to the WPWP stack could include higher resolution planktonic $\delta^{18}O$ in the older portion of the stack and better age constraints beyond 37 ka.

Analyses of the stack's standard deviation and alignment uncertainty suggest that BIGMACS performs similarly well stacking WPWP planktonic $\delta^{18}O$ as it does for regional benthic $\delta^{18}O$ data. The new stack has weaker glacial-interglacial amplitudes and orbital power for WPWP planktonic $\delta^{18}O$ change than benthic $\delta^{18}O$ stacks over the last 800 ka. WPWP planktonic $\delta^{18}O$ change is also somewhat weaker than estimated based on global ice volume and IPWP SST change, perhaps due to spatial heterogeneity or surface salinity change. Differences in glacial-interglacial amplitudes between the WPWP planktonic stack and benthic $\delta^{18}O$ stacks validate that these differences are characteristic of planktonic $\delta^{18}O$ throughout the WPWP. Furthermore, stratigraphic alignments of planktonic $\delta^{18}O$ from cores near the WPWP should produce more reliable relative age estimates when aligned to the WPWP planktonic stack instead of a benthic $\delta^{18}O$ stack.

### Author contribution
CLB curated the previously published data, performed the formal analysis, visualization, and prepared the original draft. DSR was involved with age model methodology/software development, visualization, and manuscript review and editing. LEL conceptualized the project, supervised, and contributed to the original draft preparation and manuscript review and editing. SCB provided validation and manuscript review and editing.

### Competing interests
The authors declare that they have no conflict of interest.

### Acknowledgments
This work was supported in part by the National Science Foundation (NSF) under the project number OCE-1760878.

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
