# Peer review of "An 800-kyr planktonic $\delta^{18}$ O stack for the Western Pacific Warm Pool"

_Earth System Science Data, 2023_

## Referee Comment (RC1)

The authors provide a nice western Pacific warm pool planktonic foraminifera oxygen isotope during the past 800 ka with novel stack method. This is important and timely for the further paleoceanography studies in this region. There are just few minor points I would like to ask the authors to address.

Lack of references in the introduction section

In the introduction section, the authors largely rely on only very few references (Lea et al., 2000 and Tachikawa et al., 2014) to present some basic description of western Pacific warm pool region. Although these 2 references are crucial and important, however, these are definitely not modern physical oceanography papers. If the authors would like to make the reference list be more concise, then perhaps the authors could add "and references therein" to clarify is not just these 2 references for the whole previous physical oceanography studies in the warm pool region. Or the better way, to cite some physical oceanography observational papers in the first 2 paragraphs of introduction section.

Figure 1: what is the software to make this figure? The authors only mentioned the reference for SST dataset.

Line 78-80, should be "30 m" and "20-75 m"; Also, although the original reference from Chuang et al. (2018) used "G. sacculifer". The genus name has been revised as "Trilobatus sacculifer; T. sacculifer".

Table 1: I wonder why the authors did not include Medina-Elizalde et al. (2005)'s ODP 806 data?

Also, "180-1115B" data should be "ODP 1115B", please revise it through the text, figure 1, and Tables.

Lastly, Lo et al. (2017) only report data back to 350 ka, but the dataset of MD05-2925 here is back to ~462(?) ka. Please clarify.

Lines 128-130, please describe why the authors would like to set the reservoir age as zero?

Line 150, should be "29oC"

Lines 186-187, if the authors take the uncertainty into account would the 1.3 permil significantly different from 1.7-1.8 permil glacial/interglacial changes? Please clarify.

Section 6.1, perhaps the authors could also refer to cores used in this study with both benthic and planktonic foraminifera d18O stratigraphy. For example, MD05-2925, Lo et al. (2017) several other reports in this core have reported there is no clear timing differences for the past 5-6 terminations (Liu et al., 2015, Lo et al., 2022).

The title of Figure 7 is not clear, what kind of "contributions" that the authors would like to address in this figure?

---

## Author Comment (AC2)

**Responses to Anonymous Referee #1**

**RC1:**
The authors provide a nice western Pacific warm pool planktonic foraminifera oxygen isotope during the past 800 ka with novel stack method. This is important and timely for further paleoceanography studies in this region. There are just a few minor points I would like to ask the authors to address.

Lack of references in the introduction section

In the introduction section, the authors largely rely on only very few references (Lea et al., 2000 and Tachikawa et al., 2014) to present some basic description of western Pacific warm pool region. Although these 2 references are crucial and important, however, these are definitely not modern physical oceanography papers. If the authors would like to make the reference list be more concise, then perhaps the authors could add "and references therein" to clarify is not just these 2 references for the whole previous physical oceanography studies in the warm pool region. Or the better way, to cite some physical oceanography observational papers in the first 2 paragraphs of introduction section.

We agree with the reviewer that the introduction section does not have a sufficient number of references. The revised manuscript will have additional references, including more modern oceanography papers pertaining to the region such as the following:

Broccoli, A. J.: Tropical cooling at the Last Glacial Maximum: An atmosphere–mixed layer ocean model simulation, J. Clim., 13, 951–976, https://doi.org/10.1175/1520-0442(2000)013<0951:TCATLG>2.0.CO;2, 2000.

De Deckker, P: The Indo-Pacific Warm Pool: critical to world oceanography and world climate, Geosci. Lett., 3, https://doi.org/10.1186/s40562-016-0054-3, 2016.

Lo, L., Chang, S.-P., Wei, K.-Y., Lee, S.-Y., Ou, T.-H., Chen, Y.-C., Chuang, C.-K., Mii, H.-S., Burr, G. S., Chen, M.-T., Tung, Y.-H., Tsai, M.-C., Hodell, D. A., Shen, C.-C.: Nonlinear climatic sensitivity to greenhouse gases over past 4 glacial/interglacial cycles. Sci Rep, 7 (1), 4626, https://doi.org/10.1038/s41598-017-04031-x, 2017.

Mayer, M., Haimberger L., Balmaseda M. A.: On the energy exchange between tropical ocean basins related to ENSO. J. Clim., 27, 6393–6403, https://doi.org/10.1175/JCLI-D-14-00123.1, 2014.

Neale, R., and Slingo, J.: The maritime continent and its role in the global climate: A GCM study. J. Climate, 16, 834–848, https://doi.org/10.1175/1520-0442(2003)016<0834:TMCAIR>2.0.CO;2, 2003.

Rosenthal, Y., Oppo, D. W., Linsley, B. K.: The amplitude and phasing of climate change during the last deglaciation in the Sulu Sea, Western Equatorial Pacific, Geophys. Res. Lett., 30 (8), https://doi.org/10.1029/2002GL016612, 2003.

Wefer, G. and Berger, W. H.: Isotope paleontology: Growth and composition of extant calcareous species, Marin. Geol., 100 (1), 207–248. https://doi.org/10.1016/0025-3227(91)90234-U, 1991.

Figure 1: what is the software to make this figure? The authors only mentioned the reference for SST dataset.

A description of figure 1 (core locations and SST map) has been added that will read, "Figure 1 was created with MATLAB's geoshow() function from the mapping toolbox (The Mathworks Inc., R2022b)."

Line 78-80, should be "30 m" and "20-75 m"; Also, although the original reference from Chuang et al. (2018) used "G. sacculifer". The genus name has been revised as "Trilobatus sacculifer; T. sacculifer".

Corrected.

Table 1: I wonder why the authors did not include Medina-Elizalde et al. (2005)'s ODP 806 data?

This data was overlooked during the original stack construction. The Medina-Elizade ODP 806 data has been added to a new version of the stack that will be used in the revised manuscript, with references added in Table 1, the Figure 2 caption, and the reference section.

Also, "180-1115B" data should be "ODP 1115B", please revise it through the text, figure 1, and Tables.

The core name was updated and made consistent throughout the text and figures.

Lastly, Lo et al. (2017) only report data back to 350 ka, but the dataset of MD05-2925 here is back to ~462(?) ka. Please clarify.

Additional data for MD05-2925 was used from a data set published by Lo (2021), but the citation was missing in the original version of the manuscript. The reference was added in the main text, Table 1, and the reference section.

Lines 128-130, please describe why the authors would like to set the reservoir age as zero?

The Marine20 calibration curve uses a model estimate of time-dependent global mean surface reservoir age, which is ~400 yr for the Holocene and increases to 800-1000 yr for 20-50 kyr ago. We set the reservoir age offset (ΔR) for our sites to 0 yr, meaning we did not change the reservoir age from the time-dependent Marine20 default. Additional description has been added to clarify that a reservoir age is still being used, as well as added notation for the reservoir age offset (ΔR).

Line 150, should be "29oC"

Corrected in revised manuscript.

Lines 186-187, if the authors take the uncertainty into account would the 1.3 permil significantly different from 1.7-1.8 permil glacial/interglacial changes? Please clarify.

The reviewer provides an excellent suggestion here. We will use the stacks' estimates of d18O uncertainty for each glacial and interglacial stage, to calculate an uncertainty estimate for the mean amplitudes of d18O change across TI-TV for both the planktonic and benthic stacks and include it in the revised manuscript.

Section 6.1, perhaps the authors could also refer to cores used in this study with both benthic and planktonic foraminifera d18O stratigraphy. For example, MD05-2925, Lo et al. (2017) several other reports in this core have reported there is no clear timing differences for the past 5-6 terminations (Liu et al., 2015, Lo et al., 2022).

Thank you for directing us to those studies. We have added information about planktonic/benthic age offsets from other publications as well as a qualitative comparison of planktonic and benthic records from 3 of the WPWP cores included in the new stack for which planktonic and benthic d18O data have been published (MD05-2925, ODP 1143, ODP 806). Figures comparing planktonic and benthic d18O plotted versus depth in these 3 cores will be added as supplementary material.

The title of Figure 7 is not clear, what kind of "contributions" that the authors would like to address in this figure?

The figure 7 caption has been modified to clarify that the figure shows the estimated ice volume and temperature contributions to planktonic $\delta^{18}$O.

---

## Author Comment (AC3)

**Responses to Manfred Mudelsee:**

**RC2/RC3 (comments reposted with typos corrected):**

An 800-kyr planktonic δ18O stack for the West Pacific Warm Pool

ESSD-2023-335

Review Manfred Mudelse 21 September 2023

The concept of the presented manuscript is fine: (A) stack construction for the WPWP, (B) usage of Bayesian age model algorithms, and (C) comparison of stack with other records (e.g., LR04) in terms of variability and spectral properties. However, there are flaws with data analysis and an amount of minor errors that render the current manuscript not publishable. My advice is to give authors enough time to re-submit a manuscript that overcomes data-analytical flaws and minor errors.

Data analysis

(1) Table 1

The MD97-2141 record stands out against the others in terms of temporal resolution. The manuscript should inform readers that indeed the 0.33-kyr resolution (Oppo et al. 2003) is a reasonable value since that record has a high sedimentation rate (5 to 15 cm/kyr) and also a fine sampling (1 cm). It would further be informative to study the effects of ex- or inclusion of that high-resolution record on results (variability and spectra). This can be done by repeating stack construction and variability and spectrum estimation without MD97-2141. Of course, keep MD97-2141 for calculation of the final stack.

Additional information about the sedimentation rate and sampling of MD97-2141 was added to the data section of the manuscript. For the revised version of the stack, the resolution of MD97-2141 was decreased one-fifth the original resolution (~0.55 kyr) which is closer to the other cores included in the stack (e.g., newly added core KX22-4, which has an average resolution of ~0.57 kyr [Zhang et al., 2021]). Although understanding the impact of record resolution on the stack could be interesting, we did not investigate a version of the stack with MD97-2141 fully excluded because of the high computational cost of producing each version of the stack.

The revised manuscript will read: "Published data for core MD97-2141 has an average sedimentation rate of 5-15 cm/yr and is sampled at 1 cm intervals with a mean sample spacing of 0.11 kyr (Oppo et al., 2003). However, we smoothed the data using a 5-point running mean sampled at every fifth point, which reduces its mean sample spacing to 0.55 kyr, so that this one record does not overly dominate the regional stack."

Zhang, Shuai; Yu, Zhoufei; Gong, Xun; Wang, Yue; Chang, Fengming; Lohmann, Gerrit; Qi, Yiquan; Li, Tiegang (2021): Precession cycles of the El Niño/Southern oscillation-like system controlled by Pacific

upper-ocean stratification. Communications Earth & Environment, 2(1), https://doi.org/10.1038/s43247-021-00305-5

Zhang, Shuai; Yu, Zhoufei; Gong, Xun; Wang, Yue; Chang, Fengming; Li, Tiegang (2021): Sable oxygen isotope and Mg/Ca ratios of planktonic foraminifera from KX97322-4 (KX22-4). PANGAEA, https://doi.org/10.1594/PANGAEA.939377

(2) Output resolution of WPWP stack (end of Section 4.1; Bowman et al. 2023)

The stack has 8101 data points covering the interval from 0 to 810 kyr at a constant temporal resolution of 0.1 kyr. While it is fine to present such a stack for visualization purposes, it is not OK to use it for variability or spectrum estimation because the 0.1 kyr resolution is smaller than the individual resolutions (by a factor ranging from 3.3 for core MD97-2141 up to nearly 40 for core MD97-2140). This boost-up of the sample size may lead to significantly overstated claimed statistical uncertainties (for variability or spectrum estimation). One analysis strategy to assess the effect would be to repeat variability and spectrum estimation for various other, coarser stack resolutions (say, from 0.1 kyr up to 1.0 kyr in steps of 0.1 kyr). Such a sensitivity study could make an interesting appendix for other researchers wishing to study time-resolution effects.

> We subsampled the stack to a 1 kyr resolution before performing spectral analysis, which was not stated in the methods section; that has now been clarified. In the revised manuscript, all analyses of the WPWP and North Atlantic stacks will also be calculated at 1 kyr spacing.

> Repeating the variability and spectrum estimations at incremented temporal resolutions may be interesting to some readers; however, we consider it beyond the scope of this study. The high resolution version of the stack is available for those who would like to perform their own investigation of the effect of age spacing on the variability and spectra.

(3) Spectrum estimation (Section 4.3)

> Usage of FFT is obsolete since it renders bad (in terms of estimation bias, variance, RMSE, etc.) estimates. This is known for decades (Thomson 1982, Percival and Walden 1993, Mudelsee, 2014). And since the stack is evenly spaced (at 0.1 kyr or up to 1.0 kyr resolution), one needs not invoke the Lomb-Scargle Fourier Transform (Schulz and Mudelsee 2002) but can work with Thomson's multitaper estimation (MTM), which is the method of choice here. See again the mentioned works (Thomson 1982, Percival and Walden 1993, Mudelsee, 2014) and literature cited therein. Mudelsee (2014) lists also software tools for MTM estimation in case there is need for the authors.

> For the revision, all spectral analysis will be done using a temporal resolution of 1 kyr. We also agree with the reviewer that more modern methods have statistical advantages, and we will implement the suggested Thomson's multitaper estimation method in place of the FFT results in the revised version of the manuscript, using the Matlab function pmtm(). The new method has minimal impact on our findings.

(4) Uncertainty presentation of stack

Time-varying standard deviation is certainly interesting, but I think that more about stack uncertainty can be learned from calculation of internal and external errors (and hence use weighting for stack calculation). Internal errors refer to individual records, while external errors measure the spread among the various contributing records. Individual records with smaller uncertainties should, hence, stronger contribute to the stack. Of course the challenge is to do justice to the fact that the number of records available depend on the investigated age. Details about weighting, internal and external errors can be found in the paper by Mudelsee et al. (2014), who constructed a Cenozoic δ18O stack.

This manuscript focuses on the application of existing stacking software to planktonic d18O records of the WPWP. An alternate stacking methodology would be required for us to be able to separate internal versus external errors in the stack, which is outside the scope of this study. However, more explanation of the BIGMACS stacking approach, why it is appropriate for this application, and how it weights data across sites will be added to the manuscript. One key point here is the assumption made by BIGMACS that all records in the stack are homogeneous, i.e., that they all share the same underlying signal (with allowance for site-specific shift and scale values). Under this assumption, all residuals/errors are assumed to be internal errors associated with sampling noise and measurement uncertainty. Therefore, when stacking with BIGMACS, it is important to choose records for inclusion in the stack that share the same regional influence, and our analysis of the standard deviation of the new stack suggests that the planktonic d18O records we included meet this criteria because we find a similar spread in values about the mean as two published regional benthic d18O stacks, which each only included sites that shared the same deep water mass composition.

In BIGMACS stack construction, each data point included in the stack is weighted equally because all measurements are assumed to be drawn from the same underlying distribution. Therefore, sites with higher resolution sampling provide more information/samples than sites with less data. The uncertainty of the d18O value of the stack at any point in time also includes the effects of relative age (alignment) uncertainty for that point in time in each core record, an advancement compared to the way age uncertainty was considered in the Cenozoic stack. Sites or time intervals with very noisy data have larger alignment uncertainties in BIGMACS.

The regional stacks constructed in BIGMACS are also different from the Cenozoic stacks in Mudelsee et al (2014) because the BIGMACS stacks include orbital-scale (and in some cases millennial-scale) regional climate variability and do not combine data across different oceanographic settings. The type of uncertainty information provided by BIGMACS is appropriate for our goal of characterizing the regional planktonic d18O variability of the WPWP and providing a tool to improve planktonic d18O-based age models for the region.

Although the BIGMACS stacking software provides no procedure for separating internal and external errors in the stack, we will add a column to Table 2 that reports the standard deviation of the residuals between the mean stack and each site (on its median age model and after applying its

estimated shift and scale). This will provide readers with additional insight into how similar the planktonic d18O record of each site is to regional mean.

Minor errors

I refer only to Abstract and References since already there appeared quite a number.

The revised manuscript was edited more thoroughly for phrasing and formatting mistakes.

Abstract, l. 1

The expression "different ... than" may sound strange to British ears.

Rephrased.

Abstract, l. 2

Write "greenhouse gas concentrations".

Corrected.

Abstract, l. 3

Insert a hyphen: "orbital-scale climate response".

Corrected.

Abstract, l. 6

Two commas inserted makes it more readable: "... and benthic δ18O stacks, also constructed using BIGMACS, demonstrate that ...".

Updated.

Abstract, l. 7

Insert a bit information: "Sixty-seven radiocarbon dates from the upper parts of four of the WPWP cores ...".

Additional information has been added, as well as updated values with the addition of new radiocarbon data.

Abstract, l. 11

The expression "0 - 450 ka" (with a hyphen) looks ugly. Either use an en-dash without spaces or else write "0 to 450 ka".

Updated throughout text.

References

(1) Do not capitalize (headline style) titles of listed journal articles (e.g. Huybers & Wunsch 2004 "Uncertainty estimates" ... and not "Uncertainty Estimates").

The reference titles have been corrected to match the style of ESSD in the revised manuscript.

(2) Do properly use superscripts (e.g., Imbrie et al. 1984 " ... revised chronology of the marine $\delta$18O ...", and not "$\delta$18O"; Lee et al. 2022 wrong "d18O", Lisiecki and Raymo 2005 wrong "$\delta$18O").

Corrected.

(3) Do give editor names for cited chapters from edited books (e.g., Imbrie et al. 1984).

Corrected.

References cited in review

Mudelsee M (2014) Climate Time Series Analysis: Classical Statistical and Bootstrap Methods. Second edition, Springer, Cham, Switzerland, 454 pp.

Mudelsee M, Bickert T, Lear CH, Lohmann G (2014) Cenozoic climate changes: A review based on time series analysis of marine benthic $\delta$18O records. Reviews of Geophysics 52:333—374.

Percival DB, Walden AT (1993) Spectral Analysis for Physical Applications: Multitaper and Conventional Univariate Techniques. Cambridge University Press, Cambridge, 583 pp.

Schulz M, Mudelsee M (2002) REDFIT: Estimating red-noise spectra directly from unevenly spaced paleoclimatic time series. Computers and Geosciences 28:421—426.

Thomson DJ (1982) Spectrum estimation and harmonic analysis. Proceedings of the IEEE 70:1055—1096.

---

## Author Comment (AC4)

**RC4:**

As stated in the introduction of the manuscript recent studies have advocated the development of regional $\delta 18O$ stacks to distinguish spatial differences in the timing and amplitude of $\delta 18O$ signals. Hence the contribution is timely and has the potential to provide a useful benchmark record in the Quaternary research. The manuscript is principally well-written, however there are two points in the methodology and some other specific ones which need revision before the study can be accepted for publication.

General comments:

It is not clear how was the MD97-2141 data resampled (lines 90-91): I note at this point that binning rather than smoothing and resampling would be a more adequate data processing to reduce the resolution of this record. In addition, the chosen 0.33 kyr mean sample spacing is still much finer compared to the cores from the WPWP (according to Table 1 the mean sample spacing of those cores ranges from 0.76 to 3.9 kyr). Binning to ~1ka might be more suitable to get close to the median resolution of the records representing the core area of the WPWP.

We agree there was not sufficient description of how the MD97-2141 data was resampled, and additional explanation was added to the text. We are not able to bin in the age domain because the age models change during the stack construction process. For the revised manuscript, we decreased the sampling resolution of MD97-2141 so that it now has an average sample spacing of ~0.55 kyr. This is similar to the resolution of a new core we are adding to the stack KX22-4, which has an average resolution of ~0.57 kyr (Zhang et al., 2021). We resampled by averaging non-overlapping groups of 5 adjacent d18O data points (5-point smoothing without overlap) and assigning the average d18O value to the depth of the central d18O sample. To the extent that the core is approximately evenly sampled in the depth domain, this approach is nearly the same as binning in the depth domain.

> Zhang, Shuai; Yu, Zhoufei; Gong, Xun; Wang, Yue; Chang, Fengming; Lohmann, Gerrit; Qi, Yiquan; Li, Tiegang (2021): Precession cycles of the El Niño/Southern oscillation-like system controlled by Pacific upper-ocean stratification. Communications Earth & Environment, 2(1), https://doi.org/10.1038/s43247-021-00305-5

> Zhang, Shuai; Yu, Zhoufei; Gong, Xun; Wang, Yue; Chang, Fengming; Li, Tiegang (2021): Sable oxygen isotope and Mg/Ca ratios of planktonic foraminifera from KX97322-4 (KX22-4). PANGAEA, https://doi.org/10.1594/PANGAEA.939377

According to my understanding the study applied a reservoir age offset (R) as 0+/-0.2 kyr (lines 129-130). The appropriateness of this R is debatable. I suggest checking Sarnthein et al., 2015 (DOI: https://doi.org/10.2458/azu_rc.57.17916 ). In particular, MD01-2378 was scrutinized in the study. Based on the inferred planktic reservoir ages typically >200yrs and >1kyrs during LGM (and probably in glacial conditions in general).

The Marine20 calibration curve uses a model estimate of time-dependent global mean surface reservoir age, which is ~400 yr for the Holocene and increases to 800-1000 yr for 20-50 kyr ago (Heaton et al., 2020). We set the reservoir age offset (ΔR) for our sites to 0 yr, meaning we did not change the reservoir age from the time-dependent Marine20 default. We assigned a 1-sigma uncertainty of 200 yr to the reservoir ages to account for possible changes to the reservoir age offset of the WPWP relative to the Marine20 time-dependent global mean reservoir age. Additional description has been added to clarify that a reservoir age is still being used, as well as added notation for the reservoir age offset (ΔR).

Although Sarnthein et al (2015) estimated larger reservoir ages than Marine20 for MD01-2378, we are removing that site from the stack because it is from the Timor Sea and, therefore, not strictly within the WPWP (as suggested by reviewer 4). Our choices not to apply an offset from Marine20 and to use a 200-yr uncertainty are consistent with other WPWP studies. The originally published age model for MD05-2930 used the Marine09 calibration without a reservoir age offset [Regoli et al., 2015]. Dang et al (2020) used the Marine13 calibration curve with offsets of <30 yr and an uncertainty of <100 yr for core KX21-2. Importantly, the main goal of this manuscript is to provide a record of orbital-scale variability in planktonic d18O; it is not intended to provide 1-kyr precision of absolute ages and the manuscript clearly describes the limitations of the stack's age model (e.g., lines 211-218).

Specific comments:

line 3: perhaps "greenhouse forcing" instead of "greenhouse gas"

Updated.

line 4: perhaps "covering the…" instead of "of the…"

Updated.

lines 24 to 26: Despite these are almost common knowledge some references can be needed. e.g. Wefer and Berger 1991 (https://doi.org/10.1016/0025-3227(91)90234-U) could be a pertinent reference.

Thank you for the reference suggestion, it was added in text.

line 46, 49, 226, 234, and 329: Please correct and update the citation: "(Lee and Rand et al., accepted)"

The final version of the paper has been published, the citation was updated appropriately within text and references.

line 50: "between 0-43 kyr BP" sounds strange

Updated.

line 76: the sentence sounds strange. I suggest rephrasing as follows: "Six of the cores span the last 300 to 500 kyrs, and five extend back to 750 ka."

Thank you for the suggestion, this has been rephrased within the text.

line 84: Please change to "from 450 to 800 ka"

Updated.

line 88: Ditto. Please change to "from 0.33 to 3.9 kyr"

Updated.

lines 149-150: The sentence is somehow repetitive. Please rephrase it.

Thank you for pointing this out. The sentence now reads: "We compare the amplitude of the new WPWP planktonic $\delta^{18}$O stack with a sea level (ice volume) record and a WPWP SST stack, each of which is converted to the amount of $\delta^{18}$O change they are expected to cause."

line 185: The sentence needs grammar checking.

Corrected.

line 212: Please change to "between 36 and 38 ka"

Updated.

line 214: Ditto. Please change to "between 30 and 40 ka"

Updated.

lines 235-236: I suggest replacing "our WPWP…" with "the new WPWP…"

Updated.

---

## Author Comment (AC5)

**Responses to Anonymous Referee #4**

**RC5:**

Within their manuscript: "*An 800-kyr planktonic δ18O stack for the West Pacific Warm Pool*" Christen Bowman et al. present a regional planktonic δ18O stack record that is created upon the basis of previously published δ18O records from the area by application of a novel dating and stacking software tool. The dataset might be useful for paleoceanographers in the future. Hence, I generally support its publication in ESSD, but find that the article has some flaws that should be remedied prior to publication. My concerns are outlined in more details below. The authors should also pay attention to a careful and precise wording/phrasing throughout the manuscript.

General comments:

> Choice of records: I wonder, if the records chosen to be included in the stack are representative for the WPWP. There are apparently many more regional planktonic (*G. ruber*) δ18O records from the WPWP available, which are not considered in the stack record. The authors do however include two records from the Timor Sea. Strictly speaking, this is not part of the WPWP. What are the selection criteria to include / exclude records within / from the stack? The criteria should be stated clearly in the text.

> Some planktonic d18O records from the WPWP were not included because the data did not extend past ~250 kyr. This age range criteria for core selection will be added within the text. We have also identified one new core to include, KX22-4 [Zhang et al., 2021] and more data from ODP site 806. Additionally, we are removing from the stack the Timor Sea cores SO18480-3 and MD01-2378. Collectively, these changes in the stacks' cores result in only minor changes to the stack and our estimates of glacial-interglacial amplitudes and orbital power.

> Zhang, Shuai; Yu, Zhoufei; Gong, Xun; Wang, Yue; Chang, Fengming; Lohmann, Gerrit; Qi, Yiquan; Li, Tiegang (2021): Precession cycles of the El Niño/Southern oscillation-like system controlled by Pacific upper-ocean stratification. Communications Earth & Environment, 2(1), https://doi.org/10.1038/s43247-021-00305-5

> Related to my previous point, I have another comment: Throughout the manuscript the authors refer to the "WPWP". I am wondering if it is reasonable here, because the authors include two core sites from the Timor Sea (SO18480-3, MD01-2378). Wouldn't it be more precise to refer to the Indo-Pacific Warm Pool (IPWP)? I however note that additional records from the tropical eastern Indian Ocean might be needed to cover the entire IPWP. If the authors use WPWP, shouldn't it be "Western Pacific Warm Pool" instead of "West Pacific Warm Pool" throughout the manuscript?

> The WPWP name was updated to the Western Pacific Warm Pool. We agree that the Timor Sea cores are slightly outside the bounds of the WPWP and are more heavily influenced by the Asian monsoon system. We now exclude these Timor Sea sites from the stack and we have added one new site from within the WPWP as well as additional data from ODP 806. The new stack without these Timor Sea sites is quite similar to the previous one.

Section 3 – Data: There are radiocarbon dates of cores KX21-2 (Dang et al., 2020) and MD97-2141 (Oppo et al., 2003), which should be included in this study. The KX21-2 data are presented within the original publication, the MD97-2141 data can be found here: https://www.ncei.noaa.gov/pub/data/paleo/contributions_by_author/oppo2003b/oppo2003b.txt

Thank you for directing us to the additional data, they have been included in the new version of the stack used in the revised manuscript.

Please reference the original datasets, not only the original publications. If I regard it correctly, the original records are mostly deposited online and subsequent users should be able to cite the original datasets directly.

For datasets that have separate citations available, they will be added to the revision.

Comparison of the planktonic WPWP stack to the LR04 / LS16 benthic stacks: Why don't the authors compare planktonic and benthic δ18O records of the cores they are using to create the stack to get more direct assessments of the offsets between planktonic and benthic δ18O records? Benthic δ18O records are available for at least some of the cores.

The comparison of the new planktonic d18O WPWP stack to the timing of global average benthic d18O stacks provides justification for using LR04 as the initial alignment target for the WPWP stack. In the revised manuscript we will also add discussion of possible timing differences between WPWP planktonic and benthic d18O data based on 3 cores for which both proxies have been measured (MD05-2925, ODP 1143, and ODP 806). We will also add supplementary figures showing the planktonic and benthic d18O data from these cores plotted on their shared depth scales.

The authors present a stack record with a temporal resolution / time steps of 0.1 kyr, although the resolution of the individual records that go into the stack ranges between 0.33 and 2.3 kyr. The stack thus feigns a higher resolution than given, which should be avoided. The question is how this affects the statistical analyses presented in the article.

Our stack is produced with a temporal resolution of 0.1 kyr, but for spectral analysis the stack was first sub-sampled to have a 1 kyr resolution. This has been clarified in the methods section. Regarding other types of statistical analysis, the use of Gaussian process regression correctly reflects the relative uncertainty of stack values where data are relatively sparse compared to more densely sampled portions. This is why the stack's standard deviation is narrower in the more recent time interval (e.g., 0.15‰ for 0-60 ka) than for 500-800 ka, when fewer cores/measurements are available (as discussed on lines 234-238).

The authors align their planktonic δ18O stack to the LR04 stack. They argue that there is almost no time shift between the planktonic and benthic records. Considering this, and by looking at the two stack records in comparison (Figure 4), the question arises, why scientists should use the planktonic δ18O stack

instead of the LR04 or more recent regional LS16 benthic stacks in the future. I think that should be pointed out more clearly within the article.

We agree that the manuscript needs to more clearly describe the usefulness of the new WPWP stack as a regional alignment target that will improve relative age models and alignments of other planktonic WPWP records. We want this intended purpose to be easily identifiable for readers.

The similarity in timing of d18O change between the planktonic and benthic stacks provides justification for our use of the LR04 stack as an initial alignment target. However, there are differences in the amplitude of d18O change between the planktonic and benthic stacks and local WPWP signals that make the WPWP planktonic stack a preferable regional alignment target. We will add an example to the manuscript that demonstrates how the aligned age model for core MD01-2378 (from the Timor Sea and, thus, excluded from the new version of the stack) is improved when the WPWP planktonic stack is used for the alignment target compared to using the LR04 benthic stack for alignment. We will also modify the manuscript's conclusion to clarify for which applications the WPWP planktonic stack is preferable to a benthic stack.

If the δ18O records are shifted and scaled "to better match the target stack" (line 114), how useful is it to compare amplitudes or spectral power of the WPWP and LR04 stacks?

The phrasing used here was inadvertently confusing, and the revision will explain this more clearly. (The BIGMACS "target stack" reflects planktonic values after the first iteration of alignments.) The mean and amplitude of the final WPWP stack match the mean and amplitude of the component planktonic records; therefore, comparison of the glacial cycle amplitudes and spectral power between the WPWP and LR04 are meaningful. The reported shift and scale factors indicate how much each individual record differs from the WPWP stack. (Thus, in Table 2 the shift values have an average of ~0 and the scale values average ~1.) This approach allows the amplitude of the stack to reflect variability in the common signal across sites although there are small constant offsets or amplification factors across the WPWP region.

The authors introduce that they "seek to characterize WPWP climate on orbital timescales and its differences from high-latitude climate, which can help test hypotheses about the sensitivity of the WPWP to orbital forcing, ice volume, and greenhouse gas concentration" (lines 20-22). Later on, they compare their regional planktonic stack to a regional benthic stack, to the LR04 stack and to a regional WPWP SST stack record. They however miss to draw conclusions from their results. What can be inferred from the comparisons? What is, for instance, the value of comparing spectral power and variability of the regional planktonic stack to the global benthic stack?

We intentionally omitted any interpretation of WPWP climate mechanisms based on our understanding of the aims and scope of this journal, which provides guidance that "Any interpretation of data is outside the scope of regular articles" (https://www.earth-system-science-data.net/about/aims_and_scope.html). Further guidance on data description papers says: "Although examples of data outcomes may prove necessary to demonstrate data

quality, extensive interpretations of data – i.e. detailed analysis as an author might report in a research article – remain outside the scope of this data journal. ESSD data descriptions should instead highlight and emphasize the quality, usability, and accessibility of the dataset, database, or other data product and should describe extensive carefully prepared metadata and file structures at the data repository." (https://www.earth-system-science-data.net/about/manuscript_types.html)

Therefore, we confine our discussion/conclusions to (1) the strengths and limitations of the methods used to create the stack (to provide guidance on how the data should be used for future studies), and (2) a brief description of how this stack differs from other available stacks that researchers might consider using.

In the revised version of the manuscript we will edit the conclusion so that it ends by highlighting the primary message of the publication. Specifically, we will append to the last paragraph: "Differences in the glacial-interglacial amplitudes between the WPWP planktonic stack and benthic d18O stacks validate that these differences are characteristic of planktonic d18O throughout the WPWP. Furthermore, stratigraphic alignments of WPWP planktonic d18O will be more reliable when aligned to a WPWP planktonic stack than a benthic d18O stack."

This added conclusion will be supported by a comparison of results for aligning planktonic d18O from core MD01-2378 to the WPWP planktonic stack versus the LR04 benthic stack.

More specific comments:

Line 9: It should be "a smaller glacial-interglacial amplitude".

Updated.

Lines 19-20: "Thus, climate records of the WPWP region are expected to have features which differ from many other locations on Earth." This statement is rather general and should be more precise.

We agree that this sentence should make a more specific comparison. We revise it to say "Thus, the climate records of the WPWP region are expected to have features which differ from the high-latitude climate records often used to characterize global climate change (e.g., Lisiecki & Raymo, 2005; Past Interglacials Working Group of Pages, 2016)."

Reference: Past Interglacials Working Group of PAGES (2016), Interglacials of the last 800,000 years, Rev. Geophys., 54, 162–219, doi:10.1002/2015RG000482.

Line 25: It might appear a bit old-fashioned, but don't foraminifers have tests instead of shells? Please correct.

Corrected.

Lines 24-35: Here, I clearly miss some references.

We agree with the reviewer that the introduction section does not have a sufficient amount of references. The revised manuscript will have additional references, including more modern oceanography papers pertaining to the region including the following:

Broccoli, A. J.: Tropical cooling at the Last Glacial Maximum: An atmosphere–mixed layer ocean model simulation, J. Clim., 13, 951–976, https://doi.org/10.1175/1520-0442(2000)013<0951:TCATLG>2.0.CO;2, 2000.

De Deckker, P: The Indo-Pacific Warm Pool: critical to world oceanography and world climate, Geosci. Lett., 3, https://doi.org/10.1186/s40562-016-0054-3, 2016.

Lo, L., Chang, S.-P., Wei, K.-Y., Lee, S.-Y., Ou, T.-H., Chen, Y.-C., Chuang, C.-K., Mii, H.-S., Burr, G. S., Chen, M.-T., Tung, Y.-H., Tsai, M.-C., Hodell, D. A., Shen, C.-C.: Nonlinear climatic sensitivity to greenhouse gases over past 4 glacial/interglacial cycles. Sci Rep, 7 (1), 4626, https://doi.org/10.1038/s41598-017-04031-x, 2017.

Mayer, M., Haimberger L., Balmaseda M. A.: On the energy exchange between tropical ocean basins related to ENSO. J. Clim., 27, 6393–6403, https://doi.org/10.1175/JCLI-D-14-00123.1, 2014.

Neale, R., and Slingo, J.: The maritime continent and its role in the global climate: A GCM study. J. Climate, 16, 834–848, https://doi.org/10.1175/1520-0442(2003)016<0834:TMCAIR>2.0.CO;2, 2003.

Rosenthal, Y., Oppo, D. W., Linsley, B. K.: The amplitude and phasing of climate change during the last deglaciation in the Sulu Sea, Western Equatorial Pacific, Geophys. Res. Lett., 30 (8), https://doi.org/10.1029/2002GL016612, 2003.

Line 78: There are calcification depth estimates from the study area (e.g. Hollstein et al., 2017). Why don't the authors consider these estimates? For instance, for G. ruber, the study indicates a calcification depth of 0-75 m, rather than the 30 m indicated by Wang et al. (2000).

Thank you for directing us to this additional publication. Additional information has been included within text which will read, "All but one core in the stack uses $\delta^{18}O$ values measured from the planktonic species *Globigerinoides ruber (G. ruber)* sensu stricto (s.s.), whose depth habitat in the WPWP ranges from the upper 45 m to upper 105 m of the mixed layer depending on how calcification depth is calculated (Hollstein et al., 2017). The average mixed layer depth of the WPWP is 50 m to 100 m (Locarini et al., 2018).

Line 79: Shouldn't it be *T. sacculifer*?

Updated in the revised manuscript to "One core, 180-1115B, has data from a different planktonic species, *Trilobatus sacculifer (formerly Globigerinoides sacculifer)* whose depth habitat is 20 m to 75 m or potentially as deep as 45 m to 95 m (Sadekov et al., 2009; Hollstein et al., 2017)."

Table 1: The reference of Chuang et al. (2018) is missing in the Reference list.

Corrected.

Line 119: The sentence is not complete.

Thank you for pointing out this error; "to have" should just be "have". The corrected sentence reads: "Core sites with homogeneous planktonic $\delta^{18}O$ values have shift parameters close to 0 and scale parameters close to 1."

Lines 119-121: A short explanation for the shift and scale parameters might be helpful.

Descriptions of the record properties related to the shift and scale parameters (i.e., vertical offset and amplitude) have been added to the revised manuscript. Clarification that the shift and scale are relative to the planktonic WPWP stack has also been included. The new text will read, "The shift/scale values represent how each core is adjusted optimize fit with the planktonic WPWP stack; the shift applies a constant d18O offset between the core and the stack, and the scale is a multiplicative adjustment to the amplitude of the core $\delta^{18}O$ data relative to the stack."

Lines 129-130: The authors do not apply local reservoir age corrections and assume a reservoir age standard deviation of 0.2 kyr. What is the rationale behind these assumptions?

The Marine20 calibration curve includes a time-dependent reservoir age which is used in radiocarbon data calibration (Heaton et al., 2020). We set the reservoir age offset ($\Delta R$) to 0, meaning we did not change the reservoir age from the Marine20 default of 400 yrs. Additional description has been added to clarify that a reservoir age is still being used, as well as added notation for the reservoir age offset ($\Delta R$). We assigned a 1-sigma uncertainty of 200 yr to the reservoir ages to account for possible changes to the reservoir age through time.

Our choices not to apply an offset from Marine20 and to use a 200-yr uncertainty are consistent with other WPWP studies. The originally published age model for MD05-2930 used the Marine09 calibration without a reservoir offset [Regoli et al., 2015]. Dang et al (2020) used the Marine13 calibration curve with offsets of <30 yr and an uncertainty of <100 yr for core KX21-2. Importantly, the main goal of this manuscript is to provide a record of orbital-scale variability in planktonic d18O, it is not intended to provide 1-kyr precision of absolute ages and the manuscript clearly describes the limitations of the stack's age model (e.g., lines 211-218).

Lines 135-139: If I understand it correctly, the authors choose a standard deviation of 1 kyr for the alignment of the records to MIS 3 and 4, but a standard deviation of 4 kyr for the first and last δ18O measurements of each core. I think, this needs some explanation.

We agree that the revised manuscript would benefit from additional explanation here, and we will add it to the text. Specifically, the first and last additional ages with standard deviations of 4 kyr are age estimates taken from the core's previously published age models and given to BIGMACS to help aid age model construction. The prescribed uncertainty of 4 kyr accounts for potential age model differences in the core start/end ages between the BIGMACS age model and the previously published age models. In cores MD97-2141 and ODP 1115B, tie points were added with a smaller age uncertainty of 1 kyr to provide additional guidance to the BIGMACS alignment of specific features within the records to the BIGMACS stack. We are essentially correcting the default alignment by forcing certain features in the d18O record (MIS 3 and 4) to have certain ages, so we reduce the age standard deviation to 1 kyr.

These tie points in MIS 3 and 4 are necessary largely because our initial alignment target is the LR04 benthic d18O stack, which has different amplitudes for the isotopic stages than the planktonic d18O records being aligned. The revised manuscript will include an example of an alignment error at the MIS 4/5 transition that occurs when planktonic d18O from MD01-2378 is aligned to the LR04 stack; in contrast, the correct alignment is found when the WPWP planktonic stack is used for the alignment target.

Line 150: "Present day": Here the authors could be more precise, and indicate the year they are referring to.

The meaning of "present day" has been clarified in the revised manuscript to the time interval used for the World Oceans Atlas mean annual SST for the WPWP, which is 1955 - 2018 (Locarini et al., 2018)

Section 4.3: Please indicate, which software / tool was used to perform spectral analysis.

Updated in the manuscript methods and figure captions to indicate spectral analysis was performed using MATLAB's pmtm( ) function. The revised manuscript will include this information and a citation for Matlab.

Line 185: "weaker the glacial-interglacial amplitude". Delete "the".

Corrected.

Line 254: "most negative". Please rephrase.

Updated in the revised manuscript.

Line 261: "where glacial surface water δ18Osw was more positive at the LGM". Please rephrase. I assume that the authors want to express that glacial δ18Osw was higher.

Updated in the revised manuscript.

Line 262: Which sites?

Thank you for pointing out the need to be more specific here. The revised text will read: "Previous studies show regional differences in $\delta^{18}O_{seawater}$ that may explain the reduced amplitude of planktonic $\delta^{18}O$ change at sites ODP-769A, MD97-2141, KX21-2 and ODP-806 in the Sulu Sea and eastern WPWP (de Garidel-Thoron et al., 2005)."

Line 333: It should be "which has a higher resolution".

Corrected.

Tables 1 and 2: Could the authors maybe sort the entries of these tables?

This is an excellent suggestion. Cores will be sorted by longitude (west-to-east) in both tables and in Figure 2 for the revised manuscript.

References

Dang, H., Wu, J., Xiong, Z., Qiao, P., & Li, T. (2020). Orbital and sea-level changes regulate the iron-associated sediment supplies from Papua New Guinea to the equatorial Pacific. Quaternary Science Reviews, 239. https://doi.org/10.1016/j.quascirev.2020.106361.

Hollstein, M., Mohtadi, M., Rosenthal, Y., Moffa Sanchez, P., Oppo, D., Martínez-Méndez, G., . . . Hebbeln, D. (2017). Stable oxygen isotopes and Mg/Ca in planktic foraminifera from modern surface sediments of the Western Pacific Warm Pool: Implications for thermocline reconstructions. Paleoceanography. https://doi.org/10.1002/2017PA003122.

Oppo, D. W., Linsley, B. K., Rosenthal, Y., Dannenmann, S., & Beaufort, L. (2003). Orbital and suborbital climate variability in the Sulu Sea, western tropical Pacific. Geochemistry, Geophysics, Geosystems, 4(1), 1-20. https://doi.org/10.1029/2001gc000260.

---

## Author Response (AR1)

**Overview**

We wish to thank the four reviewers for their thorough and helpful reviews of our manuscript. They are greatly appreciated, and we are confident that we have addressed all their concerns through revision. We begin by providing an overview of the reviewers' concerns, our responses, and revisions to the manuscript. For each posted reviewer's comments, we post a reply with detailed responses (in blue) to each individual comment (in black).

Most of the reviewer comments can be sorted into six areas of concern:

(1) More background information about the WPWP
    - The revised manuscript will include more references and description of the climatology of the WPWP

(2) Changes to the selection of cores to be included in the stack: Some reviewers identified WPWP cores (or 14C data) that were not included in the original stack, and concern was also expressed that two cores were from the Timor Sea, which is not strictly within the WPWP.
    - Some WPWP cores mentioned by the reviewers were excluded from our stack based on their relatively short lengths (<350 kyr). However, we have added one new WPWP core to the stack as well as additional d18O data from site 806 and 14C data from KX21-2 that were overlooked in our additional compilation. Along with adding the newly found data, we decided to exclude the two Timor Sea cores. These changes in the stack's component cores have only a very small effect on the stack and do not substantially change any of our conclusions.
    - We also describe in more detail the technique used to reduce the time/depth resolution of MD97-2141, and we change its mean sampling resolution to 0.55 kyr, which is approximately the same as the next-highest record, the newly added core KX22-4.

(3) Reviewers requested more information about the stacking methods, particularly the 14C reservoir ages, using LR04 as our alignment target, specification of the new stack's uncertainty, and the shift and scale parameters.
    - Our description of the 14C reservoir ages was not sufficiently clear. The Marine20 calibration curve includes time-dependent estimates of mean reservoir ages varying from ~400-1000 yr. We chose not to apply any additional offset from the Marine20 reservoir ages (i.e., $\Delta R = 0$) and to specify an uncertainty of 200 yr (1-sigma) in the reservoir age offset. This is approximately consistent with published 14C age models for some of the cores in the stack. However, these 14C calibration choices will only affect stack ages by ~1 kyr for the 0-40 ka portion of the stack, whereas our main objective is to provide an orbital-scale alignment target for WPWP planktonic d18O.
    - The 14C age models provide one source of evidence that WPWP planktonic d18O varied nearly synchronously with the timing of change in the LR04 benthic stack across T1. Another source of evidence, which reviewers suggested we include, is comparison of planktonic d18O and benthic d18O data measured within the same core. Three of the cores included in our stack have published planktonic and benthic d18O records, and we will include supplemental figures of these data and discuss their implications in the revised manuscript.
    - The WPWP stack was constructed with relatively new stacking software, and reviewers wanted more information about how it works. The revised version of the manuscript will include more explanation of the shift and scale parameters, clarify that the amplitude of the new stack reflects the mean amplitude of its component records, and better explain how the individual data used to construct the stack affect the calculated d18O uncertainty of the stack. We will also report the standard deviation of residuals between each core and the stack.

(4) Statistical comparison of the new WPWP stack with previously published records (e.g., spectral analysis methods)
- Spectral analysis was performed by matching the (even) sample spacing of the two records being compared (e.g., 1-kyr sampling was used for spectral analysis of both the WPWP stack and the LR04 stack); however, this was not adequately described in the methods. Based on reviewer feedback, we clarify that 1-kyr sample spacing is used for all calculated spectra. The revision also uses the multitaper method (i.e., the pmtm function in Matlab) instead of FFT.
- The revision also provides confidence intervals for the estimates of glacial-interglacial amplitudes for the WPWP planktonic stack and the benthic d18O stacks.

(5) More discussion of interpretation and conclusions, such as the suggested uses of the new stack and implications for better understanding WPWP climate dynamics.
- As this manuscript is intended to be a data description paper, our main goals are to document the methods used to construct the stack and provide guidance on appropriate uses of the stack. Therefore, we intentionally avoid discussion and conclusions about WPWP climate dynamics.
- However, the reviewers' feedback made clear that the manuscript needs more explanation of the appropriate use of the stack; it is primarily intended to provide a stratigraphic alignment target for WPWP planktonic d18O records. Stratigraphic alignment is most reliable when two records share nearly identical underlying signals (i.e., with differences attributable to noise). Therefore, systematic differences in the orbital-scale features between WPWP planktonic d18O and benthic stacks will degrade the accuracy of alignments. The revised manuscript now includes an example comparing the alignment of a WPWP planktonic d18O record to either our new planktonic stack or to the LR04 benthic stack. This example demonstrates that differences in the orbital-scale features of planktonic versus benthic stacks can affect the accuracy of the aligned age models.

(6) Typos and other copy-editing concerns
- We apologize for the numerous errors that slipped through in our initial submission and appreciate the reviewers' patience identifying them. All identified errors have been corrected, and we more carefully proof-read the revised manuscript before submission.

**Reviewer 1:**
The authors provide a nice western Pacific warm pool planktonic foraminifera oxygen isotope during the past 800 ka with novel stack method. This is important and timely for further paleoceanography studies in this region. There are just a few minor points I would like to ask the authors to address.

Lack of references in the introduction section

In the introduction section, the authors largely rely on only very few references (Lea et al., 2000 and Tachikawa et al., 2014) to present some basic description of western Pacific warm pool region. Although these 2 references are crucial and important, however, these are definitely not modern physical oceanography papers. If the authors would like to make the reference list be more concise, then perhaps the authors could add "and references therein" to clarify is not just these 2 references for the whole previous physical oceanography studies in the warm pool region. Or the better way, to cite some physical oceanography observational papers in the first 2 paragraphs of introduction section.

We agree with the reviewer that the introduction section does not have a sufficient number of references. The revised manuscript will have additional references, including more modern oceanography papers pertaining to the region. The new text (lines 14-36) reads:

"The tropical Pacific is an important source of heat and moisture to the atmosphere (e.g., De Deckker, 2016; Neale and Slingo, 2003; Mayer et al., 2014) and is thought to have a strong impact on global climate responses during glacial cycles (Lea et al., 2000). Prior studies suggest that the climate of the Western Pacific Warm Pool (WPWP), which is defined by mean annual sea surface temperatures (SST) above 28°C, responds primarily to changes in greenhouse gas concentrations due to the region's large distance from high-latitude ice sheets (Broccoli et al., 2000; Lea, 2004; Tachikawa et al., 2014). Additionally, Earth's orbital cycles cause seasonal variations in insolation or incoming solar radiation, which affect Earth's high and low latitudes differently. In the WPPW, only 0.3°C of SST change is attributed to orbital forcing during the Late Pleistocene (Tachikawa et al., 2014). Thus, climate records of the WPWP region are expected to have features which differ from the high-latitude climate records often used to describe global climate change (e.g., Lisiecki & Raymo, 2005; Past Interglacials Working Group of PAGES, 2016). Here we seek to characterize WPWP climate on orbital timescales and its differences from high-latitude climate, which can help test hypotheses about the sensitivity of the WPWP to orbital forcing, ice volume, and greenhouse gas concentration.

One of the most commonly used paleoceanographic climate proxies is the ratio of oxygen isotopes, denoted as $\delta^{18}O$, in calcium carbonate from foraminiferal tests; this proxy is affected by both water temperature and the $\delta^{18}O$ of seawater, which varies with global ice volume as well as local salinity (Wefer and Berger, 1991). The two general types of foraminifera are benthic and planktonic, which live in the deep ocean and surface ocean, respectively. Benthic $\delta^{18}O$ is considered a high-latitude climate proxy because deep water temperature is set in high-latitude deep water formation regions and because global ice volume responds primarily to high-latitude northern hemisphere summer insolation. However, planktonic $\delta^{18}O$ is influenced by both high-latitude ice volume and local SST and salinity (Rosenthal et al., 2003). Previous studies from the WPWP have shown smaller glacial-interglacial amplitudes of planktonic $\delta^{18}O$ change than in benthic $\delta^{18}O$ or planktonic $\delta^{18}O$ from other regions (Lea et al., 2000; de Garidel-Thoron et al., 2005). This difference has been attributed to smaller sea surface temperature fluctuations and salinity changes in the WPWP (Broccoli et al., 2000; Lea et al., 2000; de Garidel-Thoron et al., 2005). Here we present a stack (time-dependent average) of planktonic $\delta^{18}O$ records from ten cores across the WPWP to provide a record of its regional responses over the past 800 ka, which can be compared to the high-latitude response of global and regional benthic $\delta^{18}O$ stacks."

New citations included in the introduction are:

Broccoli, A. J.: Tropical cooling at the Last Glacial Maximum: An atmosphere–mixed layer ocean model simulation, J. Clim., 13, 951–976, https://doi.org/10.1175/1520-0442(2000)013<0951:TCATLG>2.0.CO;2, 2000.

De Deckker, P: The Indo-Pacific Warm Pool: critical to world oceanography and world climate, Geosci. Lett., 3, https://doi.org/10.1186/s40562-016-0054-3, 2016.

Lo, L., Chang, S.-P., Wei, K.-Y., Lee, S.-Y., Ou, T.-H., Chen, Y.-C., Chuang, C.-K., Mii, H.-S., Burr, G. S., Chen, M.-T., Tung, Y.-H., Tsai, M.-C., Hodell, D. A., Shen, C.-C.: Nonlinear climatic sensitivity to greenhouse

gases over past 4 glacial/interglacial cycles. Sci Rep, 7 (1), 4626, https://doi.org/10.1038/s41598-017-04031-x, 2017.

Mayer, M., Haimberger L., Balmaseda M. A.: On the energy exchange between tropical ocean basins related to ENSO. J. Clim., 27, 6393–6403, https://doi.org/10.1175/JCLI-D-14-00123.1, 2014.

Neale, R., and Slingo, J.: The maritime continent and its role in the global climate: A GCM study. J. Climate, 16, 834–848, https://doi.org/10.1175/1520-0442(2003)016<0834:TMCAIR>2.0.CO;2, 2003.

Rosenthal, Y., Oppo, D. W., Linsley, B. K.: The amplitude and phasing of climate change during the last deglaciation in the Sulu Sea, Western Equatorial Pacific, Geophys. Res. Lett., 30 (8), https://doi.org/10.1029/2002GL016612, 2003.

Wefer, G. and Berger, W. H.: Isotope paleontology: Growth and composition of extant calcareous species, Marin. Geol., 100 (1), 207–248. https://doi.org/10.1016/0025-3227(91)90234-U, 1991.

Figure 1: what is the software to make this figure? The authors only mentioned the reference for SST dataset.

The Figure 1 caption now includes the statement, "Figure 1 was created with MATLAB's geoshow() function from the mapping toolbox (The Mathworks Inc., R2023a)."

Line 78-80, should be "30 m" and "20-75 m"; Also, although the original reference from Chuang et al. (2018) used "G. sacculifer". The genus name has been revised as "Trilobatus sacculifer; T. sacculifer".

Corrected. The revised manuscript reads, "One core, ODP 1115B, has data from a different planktonic species, *Trilobatus sacculifer* (formerly *Globigerinoides sacculifer*)..." (lines 89-91)

Table 1: I wonder why the authors did not include Medina-Elizalde et al. (2005)'s ODP 806 data?

This data was overlooked during the original stack construction. The Medina-Elizade ODP 806 data has been added to the new version of the stack used in the revised manuscript, with references added in the Figure 2 caption, the reference section. Reference added:

Medina-Elizalde, M., and Lea, D. W.: (Table S2) Stable oxygen isotope record and Mg/Ca ratios of Globigerinoides ruber from ODP Hole 130-806B. PANGAEA [data set], https://doi.org/10.1594/PANGAEA.772014, 2005.

Also, "180-1115B" data should be "ODP 1115B", please revise it through the text, figure 1, and Tables.

The core name was updated to ODP-1115B and made consistent throughout the text and figures.

Lastly, Lo et al. (2017) only report data back to 350 ka, but the dataset of MD05-2925 here is back to ~462(?) ka. Please clarify.

Additional data for MD05-2925 was used from a data set published by Lo (2021), but the citation was missing in the original version of the manuscript. The reference was added in the main text, Table 1, and the reference section.

Lo, L.: A dataset of the Mid-Brunhes period at site MD05-2925, Solomon Sea: Surface-subsurface planktonic foraminifera stable oxygen isotope and Mg/Ca ratios, Mendeley Data [data set], https://data.mendeley.com/datasets/9c2nnpchdh/1, 2021.

Lines 128-130, please describe why the authors would like to set the reservoir age as zero?

The Marine20 calibration curve uses a model estimate of time-dependent global mean surface reservoir age, which is ~400 yr for the Holocene and increases to 800-1000 yr for 20-50 kyr ago. We set the reservoir age offset (ΔR) for our sites to 0 yr, meaning we did not change the reservoir age from the time-dependent Marine20 default. Additional description has been added to clarify that a reservoir age is still being used, as well as added notation for the reservoir age offset (ΔR).

The new manuscript reads (lines 154 - 159): "We calibrated radiocarbon ages using the Marine20 calibration curve, which uses a model estimate of time-dependent global mean surface reservoir age, with values of ~400 yr in the Holocene and 800 to 1000 yr from 20 to 50 ka (Heaton et al., 2020). We set the reservoir age offset (ΔR) for our sites to 0 yr, meaning we did not change the reservoir age from the time-dependent Marine20 default."

Line 150, should be "29oC"

Corrected in revised manuscript. (line 182)

Lines 186-187, if the authors take the uncertainty into account would the 1.3 permil significantly different from 1.7-1.8 permil glacial/interglacial changes? Please clarify.

The reviewer provides an excellent suggestion here. In the revised manuscript we use the stacks' estimates of d18O uncertainty for each glacial and interglacial stage, to calculate an uncertainty estimate for the mean amplitudes of d18O change across TI-TV for both the planktonic and benthic stacks.

The revised manuscript reads (lines 220-223): "The average glacial/interglacial amplitude for Terminations I to V is 1.7 ± 0.1 ‰ and 1.8 ± 0.1 ‰ in the LR04 and North Atlantic benthic stacks, respectively, but only 1.2 ± 0.1 ‰ in the WPWP planktonic stack. (One standard deviation uncertainty for the mean amplitude of each stack is calculated using the time-dependent standard deviation of $\delta^{18}O$ in each stack.)"

Section 6.1, perhaps the authors could also refer to cores used in this study with both benthic and planktonic foraminifera d18O stratigraphy. For example, MD05-2925, Lo et al. (2017) several other reports in this core have reported there is no clear timing differences for the past 5-6 terminations (Liu et al., 2015, Lo et al., 2022).

Thank you for directing us to those studies. We have added information about planktonic/benthic age offsets from other publications as well as a qualitative comparison of planktonic and benthic records from 3 of the WPWP cores included in the new stack for which planktonic and benthic d18O data have been published (MD05-2925, ODP 1143, ODP 806). Figures comparing planktonic and benthic d18O plotted versus depth in these 3 cores have been  added as supplementary material.

The following has been added to the revised manuscript (lines 244-248): "We also compare changes in planktonic and benthic $\delta^{18}$O measured within individual WPWP cores as a function of depth for MD05-2925, ODP-1143, and ODP-806 (Lo et al., 2019; Lo, 2021; Tian et al., 2006; Medina-Elizalde and Lea, 2005; Bickert et al., 1993). These cores do not show a consistent lead/lag between the planktonic $\delta^{18}$O and benthic $\delta^{18}$O records (**Fig. S1 - S3**), additionally indicating that the timing of WPWP planktonic and benthic $\delta^{18}$O change is similar on orbital time scales."

The title of Figure 7 is not clear, what kind of "contributions" that the authors would like to address in this figure?

That figure title (Figure 8 in the revised manuscript) has been modified to clarify that the figure shows the estimated ice volume and temperature contributions to planktonic $\delta^{18}$O. The caption in the revised manuscript reads: "Figure 8. Ice volume and temperature contributions to WPWP planktonic $\delta^{18}$O."

**Reviewer 2:**

An 800-kyr planktonic δ18O stack for the West Pacific Warm Pool

ESSD-2023-335

Review Manfred Mudelse 21 September 2023

The concept of the presented manuscript is fine: (A) stack construction for the WPWP, (B) usage of Bayesian age model algorithms, and (C) comparison of stack with other records (e.g., LR04) in terms of variability and spectral properties. However, there are flaws with data analysis and an amount of minor errors that render the current manuscript not publishable. My advice is to give authors enough time to re-submit a manuscript that overcomes data-analytical flaws and minor errors.

Data analysis

(1) Table 1

The MD97-2141 record stands out against the others in terms of temporal resolution. The manuscript should inform readers that indeed the 0.33-kyr resolution (Oppo et al. 2003) is a reasonable value since that record has a high sedimentation rate (5 to 15 cm/kyr) and also a fine sampling (1 cm). It would further be informative to study the effects of ex- or inclusion of that high-resolution record on results (variability and spectra). This can be done by repeating stack construction and variability and spectrum estimation without MD97-2141. Of course, keep MD97-2141 for calculation of the final stack.

Additional information about the sedimentation rate and sampling of MD97-2141 was added to the data section of the manuscript. For the revised version of the stack, the resolution of MD97-2141 was decreased one-fifth the original resolution (~0.55 kyr) which is closer to the other cores included in the stack (e.g., newly added core KX22-4, which has an average resolution of ~0.57 kyr [Zhang et al., 2021]). Although understanding the impact of record resolution on the stack could be interesting, we did not investigate a version of the stack with MD97-2141 fully excluded because of the high computational cost of producing each version of the stack.

Zhang, Shuai; Yu, Zhoufei; Gong, Xun; Wang, Yue; Chang, Fengming; Li, Tiegang (2021): Sable oxygen isotope and Mg/Ca ratios of planktonic foraminifera from KX97322-4 (KX22-4). PANGAEA, https://doi.org/10.1594/PANGAEA.939377

The revised manuscript reads (lines 102-105): "Published data for core MD97-2141 has an average sedimentation rate of 5-15 cm/yr and is sampled at 1 cm intervals with a mean sample spacing of 0.11 kyr (Oppo et al., 2003). However, we smoothed the data using a 5-point running mean sampled at every fifth point, which reduces its mean sample spacing to 0.55 kyr, so that this one record does not overly dominate the regional stack."

(2) Output resolution of WPWP stack (end of Section 4.1; Bowman et al. 2023)

The stack has 8101 data points covering the interval from 0 to 810 kyr at a constant temporal resolution of 0.1 kyr. While it is fine to present such a stack for visualization purposes, it is not OK to use it for variability or spectrum estimation because the 0.1 kyr resolution is smaller than the individual resolutions (by a factor ranging from 3.3 for core MD97-2141 up to nearly 40 for core MD97-2140). This boost-up of the sample size may lead to significantly overstated claimed statistical uncertainties (for variability or spectrum estimation). One analysis strategy to assess the effect would be to repeat variability and spectrum estimation for various other, coarser stack resolutions (say, from 0.1 kyr up to 1.0 kyr in steps of 0.1 kyr). Such a sensitivity study could make an interesting appendix for other researchers wishing to study time-resolution effects.

We subsampled the stack to a 1 kyr resolution before performing spectral analysis, which was not stated in the methods section; that has now been clarified. The revised manuscript reads (line 186): "Both stacks were sub-sampled at 1 kyr spacing from 0 to 800 ka…"

The Gaussian process regression used to create the stacks accounts for the resolution of the data and generates larger estimates for uncertainty where the data are sparse. Because stack uncertainty estimates can be generated as a time-continuous function, we used the original 0.1 kyr resolution of the stacks to calculate the mean uncertainty of the stacks.

Repeating the variability and spectrum estimations at incremented temporal resolutions may be interesting to some readers; however, we consider it beyond the scope of this study. The high resolution version of the stack is available for those who would like to perform their own investigation of the effect of age spacing on the variability and spectra.

(3) Spectrum estimation (Section 4.3)

Usage of FFT is obsolete since it renders bad (in terms of estimation bias, variance, RMSE, etc.) estimates. This is known for decades (Thomson 1982, Percival and Walden 1993, Mudelsee, 2014). And since the stack is evenly spaced (at 0.1 kyr or up to 1.0 kyr resolution), one needs not invoke the Lomb-Scargle Fourier Transform (Schulz and Mudelsee 2002) but can work with Thomson's multitaper estimation (MTM), which is the method of choice here. See again the mentioned works (Thomson 1982, Percival and Walden 1993, Mudelsee, 2014) and literature cited therein. Mudelsee (2014) lists also software tools for MTM estimation in case there is need for the authors.

For the revision, all spectral analysis is done using a temporal resolution of 1 kyr. We also agree with the reviewer that more modern methods have statistical advantages, and we implement the suggested

Thomson's multitaper estimation method in place of the FFT results in the revised version of the manuscript, using the Matlab function pmtm(). The new method has minimal impact on our findings.

Figures 5 and 9 have been updated with the new spectral results and the revised manuscript reads (lines 186-188): "Both stacks were sub-sampled at 1 kyr spacing from 0 to 800 ka, and power spectral density was calculated using the multitaper power spectral density estimate function pmtm( ) in MATLAB, with the number of tapers set to two, a rate of one sample per kyr, and an nfft of 512 (The MathWorks Inc., R2023a)."

(4) Uncertainty presentation of stack

Time-varying standard deviation is certainly interesting, but I think that more about stack uncertainty can be learned from calculation of internal and external errors (and hence use weighting for stack calculation). Internal errors refer to individual records, while external errors measure the spread among the various contributing records. Individual records with smaller uncertainties should, hence, stronger contribute to the stack. Of course the challenge is to do justice to the fact that the number of records available depend on the investigated age. Details about weighting, internal and external errors can be found in the paper by Mudelsee et al. (2014), who constructed a Cenozoic δ18O stack.

This manuscript focuses on the application of existing stacking software to planktonic d18O records of the WPWP. An alternate stacking methodology would be required for us to be able to separate internal versus external errors in the stack, which is outside the scope of this study. However, more explanation of the BIGMACS stacking approach, why it is appropriate for this application, and how it weights data across sites will be added to the manuscript. One key point here is the assumption made by BIGMACS that all records in the stack are homogeneous, i.e., that they all share the same underlying signal (with allowance for site-specific shift and scale values). Under this assumption, all residuals/errors are assumed to be internal errors associated with sampling noise and measurement uncertainty. Therefore, when stacking with BIGMACS, it is important to choose records for inclusion in the stack that share the same regional influence, and our analysis of the standard deviation of the new stack suggests that the planktonic d18O records we included meet this criteria because we find a similar spread in values about the mean as two published regional benthic d18O stacks, which each only included sites that shared the same deep water mass composition.

In BIGMACS stack construction, each data point included in the stack is weighted equally because all measurements are assumed to be drawn from the same underlying distribution. Therefore, sites with higher resolution sampling provide more information/samples than sites with less data. The uncertainty of the d18O value of the stack at any point in time also includes the effects of relative age (alignment) uncertainty for that point in time in each core record, an advancement compared to the way age uncertainty was considered in the Cenozoic stack. Sites or time intervals with very noisy data have larger alignment uncertainties in BIGMACS.

The regional stacks constructed in BIGMACS are also different from the Cenozoic stacks in Mudelsee et al (2014) because the BIGMACS stacks include orbital-scale (and in some cases millennial-scale) regional climate variability and do not combine data across different oceanographic settings. The type of uncertainty information provided by BIGMACS is appropriate for our goal of characterizing the regional planktonic d18O variability of the WPWP and providing a tool to improve planktonic d18O-based age models for the region.

Although the BIGMACS stacking software provides no procedure for separating internal and external errors in the stack, we will add a column to Table 2 that reports the standard deviation of the residuals between the mean stack and each site (on its median age model and after applying its estimated shift and scale). This will provide readers with additional insight into how similar the planktonic d18O record of each site is to regional mean.

The following has been added to/edited in the revised manuscript which reads (lines 145-154):

"Importantly, BIGMACS assumes that all records in the stack are homogeneous, i.e., that they all share the same underlying signal (with allowance for site-specific shift and scale values). Under this assumption, all residuals between individual $\delta^{18}$O measurements and the stack are assumed to reflect variability associated with sampling noise, measurement uncertainty and/or alignment uncertainty. Therefore, when stacking with BIGMACS, it is important to choose records for inclusion in the stack that share the same regional influence. Additionally, because all measurements are treated equally, cores with higher resolution data are more strongly weighted in the stack construction. The stack uncertainty reported by BIGMACS is the time-dependent standard deviation of a Gaussian fit to the $\delta^{18}$O residuals. To evaluate whether the assumption of homogeneity used by BIGMACS for stack construction is applicable to the WPWP planktonic $\delta^{18}$O records in our new stack, Section 6.2 compares the WPWP planktonic stack uncertainty and the average alignment uncertainty of the stacked records to results from previously published regional benthic $\delta^{18}$O stacks."

We added a column to Table 2 that shows the standard deviation of residuals between the stack and the d18O values from each aligned record (after applying BIGMACS's shift and scale estimates).

(lines 269-272): "Based on the BIGMACS assumption of homogeneity across aligned records, all $\delta^{18}$O residuals are assumed to be internal errors associated with sampling noise and measurement uncertainty and, thus, all residuals contribute similarly to estimating the stack's time-dependent standard deviation."

Minor errors

I refer only to Abstract and References since already there appeared quite a number.

The revised manuscript was edited more thoroughly for phrasing and formatting mistakes.

Abstract, l. 1

The expression "different ... than" may sound strange to British ears.

Rephrased, to (line 1): "...compared to…"

Abstract, l. 2

Write "greenhouse gas concentrations".

Revised to (line 2): "greenhouse forcing"

Abstract, l. 3

Insert a hyphen: "orbital-scale climate response".

Corrected (line 11)

Abstract, l. 6

Two commas inserted makes it more readable: "... and benthic δ18O stacks, also constructed using BIGMACS, demonstrate that ...".

Updated (line 6)

Abstract, l. 7

Insert a bit information: "Sixty-seven radiocarbon dates from the upper parts of four of the WPWP cores ...".

Additional information has been added, as well as updated values with the addition of new radiocarbon data. The revised manuscript reads (lines 7 - 9): "Sixty-five radiocarbon dates from the upper portion of five of the WPWP cores suggest that WPWP planktonic $\delta^{18}O$ change is nearly synchronous with global benthic $\delta^{18}O$ during the last glacial termination."

Abstract, l. 11

The expression "0 - 450 ka" (with a hyphen) looks ugly. Either use an en-dash without spaces or else write "0 to 450 ka".

Updated throughout text.


**Reviewer 3:**

As stated in the introduction of the manuscript recent studies have advocated the development of regional δ18O stacks to distinguish spatial differences in the timing and amplitude of δ18O signals. Hence the contribution is timely and has the potential to provide a useful benchmark record in the Quaternary research. The manuscript is principally well-written, however there are two points in the methodology and some other specific ones which need revision before the study can be accepted for publication.

General comments:

It is not clear how was the MD97-2141 data resampled (lines 90-91): I note at this point that binning rather than smoothing and resampling would be a more adequate data processing to reduce the resolution of this record. In addition, the chosen 0.33 kyr mean sample spacing is still much finer compared to the cores from the WPWP (according to Table 1 the mean sample spacing of those cores ranges from 0.76 to 3.9 kyr). Binning to ~1ka might be more suitable to get close to the median resolution of the records representing the core area of the WPWP.

We agree there was not sufficient description of how the MD97-2141 data was resampled, and additional explanation has been added to the text. We are not able to bin in the age domain because the age models change during the stack construction process. For the revised manuscript, we decreased the sampling resolution of MD97-2141 so that it now has an average sample spacing of ~0.55 kyr. This is similar to the resolution of a new core we are adding to the stack KX22-4, which has an average resolution of ~0.57 kyr (Zhang et al., 2021). We resampled by averaging non-overlapping groups of 5 adjacent d18O data points (5-point smoothing without overlap) and assigning the average d18O value to the depth of the central d18O sample. To the extent that the core is approximately evenly sampled in the depth domain, this approach is nearly the same as binning in the depth domain.

The revised manuscript reads (lines 102-105): "Published data for core MD97-2141 has an average sedimentation rate of 5-15 cm/yr and is sampled at 1 cm intervals with a mean sample spacing of 0.11 kyr (Oppo et al., 2003). However, we smoothed the data using a 5-point running mean sampled at every fifth point, which reduces its mean sample spacing to 0.55 kyr, so that this one record does not overly dominate the regional stack."

Zhang, S., Yu, Z., Gong, X., Wang, Y., Chang, F., and Li, T.: Sable oxygen isotope and Mg/Ca ratios of planktonic foraminifera from KX97322-4 (KX22-4), PANGAEA [data set], https://doi.org/10.1594/PANGAEA.939377, 2021.

According to my understanding the study applied a reservoir age offset (R) as 0+/-0.2 kyr (lines 129-130). The appropriateness of this R is debatable. I suggest checking Sarnthein et al., 2015 (DOI: https://doi.org/10.2458/azu_rc.57.17916 ). In particular, MD01-2378 was scrutinized in the study. Based on the inferred planktic reservoir ages typically >200yrs and >1kyrs during LGM (and probably in glacial conditions in general).

The Marine20 calibration curve uses a model estimate of time-dependent global mean surface reservoir age, with values of ~400 yr for the Holocene and increases to 800-1000 yr for 20-50 kyr ago (Heaton et al., 2020). We set the reservoir age offset (ΔR) for our sites to 0 yr, meaning we did not change the reservoir age from the time-dependent Marine20 default. We assigned a 1-sigma uncertainty of 200 yr to the reservoir ages to account for possible changes to the reservoir age offset of the WPWP relative to the Marine20 time-dependent global mean reservoir age. Additional description has been added to clarify that a reservoir age is still being used, as well as added notation for the reservoir age offset (ΔR).

The new manuscript reads (lines 156 - 159): "We calibrated radiocarbon ages using the Marine20 calibration curve, which uses a model estimate of time-dependent global mean surface reservoir age, with values of ~400 yr in the Holocene and 800 to 1000 yr from 20 to 50 ka (Heaton et al., 2020). We set the reservoir age offset (ΔR) for our sites to 0 yr, meaning we did not change the reservoir age from the time-dependent Marine20 default."

Although Sarnthein et al (2015) estimated larger reservoir ages than Marine20 for MD01-2378, we are removing that site from the stack because it is from the Timor Sea and, therefore, not strictly within the WPWP (as suggested by reviewer 4). Our choices not to apply an offset from Marine20 and to use a 200-yr uncertainty are consistent with other WPWP studies. The originally published age model for MD05-2930 used the Marine09 calibration without a reservoir age offset [Regoli et al., 2015]. Dang et al (2020) used the Marine13 calibration curve with offsets of <30 yr and an uncertainty of <100 yr for core KX21-2. Importantly, the main goal of this manuscript

is to provide a record of orbital-scale variability in planktonic d18O; it is not intended to provide 1-kyr precision of absolute ages and the manuscript clearly describes the limitations of the stack's age model (e.g., lines 211-218).

The revised manuscript reads (lines 250-251, and 254-256): "The relative timing of millennial-scale variability between the WPWP planktonic stack and benthic $\delta^{18}$O is more difficult to evaluate." and "The portions of our WPWP stack older than 37 ka, which are not constrained by radiocarbon data, inherit the +/- 4 kyr age uncertainty of the LR04 stack used as the initial alignment target."

The revised manuscript reads (lines 320-322): "Although the new WPWP planktonic stack can improve estimates of relative age regionally, we caution that its absolute ages are susceptible to our assumption of synchronous change in benthic $\delta^{18}$O and WPWP planktonic $\delta^{18}$O and the absolute age uncertainty of the LR04 stack."

Specific comments:

line 3: perhaps "greenhouse forcing" instead of "greenhouse gas"

Updated line 2: "...its sea surface temperatures are thought to respond primarily to changes in greenhouse forcing."

line 4: perhaps "covering the…" instead of "of the…"

Updated line 2-3: "To better characterize the orbital-scale climate response covering the WPWP,..."

lines 24 to 26: Despite these are almost common knowledge some references can be needed. e.g. Wefer and Berger 1991 (https://doi.org/10.1016/0025-3227(91)90234-U) could be a pertinent reference.

Thank you for the reference suggestion, it was added in text. The revised manuscript reads (lines 26-29):

"One of the most commonly used paleoceanographic climate proxies is the ratio of oxygen isotopes, denoted as $\delta^{18}$O, in calcium carbonate from foraminiferal tests; this proxy is affected by both water temperature and the $\delta^{18}$O of seawater, which varies with global ice volume as well as local salinity (Wefer and Berger, 1991)."

line 46, 49, 226, 234, and 329: Please correct and update the citation: "(Lee and Rand et al., accepted)"

The final version of the paper has been published, the citation was updated appropriately within text to "(Lee and Rand et al., 2023)" and references.

Lee, T., Rand, D., Lisiecki, L. E., Gebbie, G., and Lawrence, C.: Bayesian age models and stacks: combining age inferences from radiocarbon and benthic $\delta^{18}$O stratigraphic alignment, Clim. Past, 19, 1993–2012, https://doi.org/10.5194/cp-19-1993-2023, 2023.

line 50: "between 0-43 kyr BP" sounds strange

The revised manuscript reads (lines 53-54): "The new stack consists of previously published planktonic $\delta^{18}$O data and 65 radiocarbon dates ranging from 1.5 to 36.9 ka from ten cores within the WPWP."

line 76: the sentence sounds strange. I suggest rephrasing as follows: "Six of the cores span the last 300 to 500 kyrs, and five extend back to 750 ka."

Thank you for the suggestion, the revised manuscript (updated based on the new 10 core stack) reads (lines 86-87): "Four cores span the last 350 to 500 ka, and six extend back to at least 750 ka."

line 84: Please change to "from 450 to 800 ka"

The dash between numerics has been replaced throughout the revised manuscript from "-" to "to"

line 88: Ditto. Please change to "from 0.33 to 3.9 kyr"

Updated (line 101).

lines 149-150: The sentence is somehow repetitive. Please rephrase it.

Thank you for pointing this out. The revised manuscript reads (lines 173-174): "We compare the amplitude of the new WPWP planktonic $\delta^{18}$O stack with a sea level (ice volume) record and an IPWP SST stack, each of which is converted to the amount of planktonic $\delta^{18}$O change they are expected to produce."

line 185: The sentence needs grammar checking.

Corrected, the revised manuscript reads (now lines 219-220): The WPWP planktonic stack has a weaker glacial-interglacial amplitude than the global LR04 benthic $\delta^{18}$O stack (Lisiecki and Raymo, 2005) or a North Atlantic benthic $\delta^{18}$O stack produced by BIGMACS (Hobart et al., 2023)."

line 212: Please change to "between 36 and 38 ka"

Updated, line 251: "Apparent differences in timing of a millennial-scale feature in the stacks between 36 and 38 ka …"

line 214: Ditto. Please change to "between 30 and 40 ka"

Updated, line 253: "...between 30 and 40 ka."

lines 235-236: I suggest replacing "our WPWP…" with "the new WPWP…"

Updated, line 276: "...the new WPWP planktonic stack has a mean standard deviation of 0.16 ‰ for the same age range."

**Reviewer #4:**

Within their manuscript: "*An 800-kyr planktonic δ18O stack for the West Pacific Warm Pool*" Christen Bowman et al. present a regional planktonic δ18O stack record that is created upon the basis of previously published δ18O records from the area by application of a novel dating and stacking software tool. The dataset might be useful for

paleoceanographers in the future. Hence, I generally support its publication in ESSD, but find that the article has some flaws that should be remedied prior to publication. My concerns are outlined in more details below. The authors should also pay attention to a careful and precise wording/phrasing throughout the manuscript.

General comments:

Choice of records: I wonder, if the records chosen to be included in the stack are representative for the WPWP. There are apparently many more regional planktonic (*G. ruber*) δ18O records from the WPWP available, which are not considered in the stack record. The authors do however include two records from the Timor Sea. Strictly speaking, this is not part of the WPWP. What are the selection criteria to include / exclude records within / from the stack? The criteria should be stated clearly in the text.

Some planktonic d18O records from the WPWP were not included because the data did not extend past ~250 kyr. This age range criteria for core selection will be added within the text. We have also identified one new core to include, KX22-4 (Zhang et al., 2021) and more data from ODP site 806 (Medina-Elizalde and Lea, 2005). Additionally, we are removing from the stack the Timor Sea cores SO18480-3 and MD01-2378. Collectively, these changes in the stacks' cores result in only minor changes to the stack and our estimates of glacial-interglacial amplitudes and orbital power.

The revised manuscript reads (lines 85-86): " Cores were included in the stack based on their location in the WPWP, an age range spanning at least three glacial cycles, and an average time resolution of at least 4 kyr."

References added:

Medina-Elizalde, M., and Lea, D. W.: (Table S2) Stable oxygen isotope record and Mg/Ca ratios of Globigerinoides ruber from ODP Hole 130-806B. PANGAEA [data set], https://doi.org/10.1594/PANGAEA.772014, 2005.

Zhang, S., Yu, Z., Gong, X., Wang, Y., Chang, F., and Li, T.: Sable oxygen isotope and Mg/Ca ratios of planktonic foraminifera from KX97322-4 (KX22-4), PANGAEA [data set], https://doi.org/10.1594/PANGAEA.939377, 2021.

Related to my previous point, I have another comment: Throughout the manuscript the authors refer to the "WPWP". I am wondering if it is reasonable here, because the authors include two core sites from the Timor Sea (SO18480-3, MD01-2378). Wouldn't it be more precise to refer to the Indo-Pacific Warm Pool (IPWP)? I however note that additional records from the tropical eastern Indian Ocean might be needed to cover the entire IPWP. If the authors use WPWP, shouldn't it be "Western Pacific Warm Pool" instead of "West Pacific Warm Pool" throughout the manuscript?

The WPWP name was updated to the Western Pacific Warm Pool. We agree that the Timor Sea cores are slightly outside the bounds of the WPWP and are more heavily influenced by the Asian monsoon system. We now exclude these Timor Sea sites from the stack and we have added one new site from within the WPWP as well as additional data from ODP 806. The new stack without these Timor Sea sites is quite similar to the previous one.

The revised manuscript now reads (title and throughout the text): "Western Pacific Warm Pool"

Section 3 – Data: There are radiocarbon dates of cores KX21-2 (Dang et al., 2020) and MD97-2141 (Oppo et al., 2003), which should be included in this study. The KX21-2 data are presented within the original publication, the MD97-2141 data can be found here:
https://www.ncei.noaa.gov/pub/data/paleo/contributions_by_author/oppo2003b/oppo2003b.txt

Thank you for directing us to the additional data, they have been included along with data from the newly added core KX22-4 in the new version of the stack for the revised manuscript. Conversely, some previously used 14C dates have been removed due to the exclusion of the two Timor Sea sites.

Lines 105-107: "Additionally, we constrain the stack age model using 65 previously published radiocarbon measurements ranging from 1.52 ka to 36.9 ka from five cores (Oppo et al., 2003; Regoli et al., 2015; Lo et al., 2017, Dang et al., 2020, Zhang et al., 2021)."

Please reference the original datasets, not only the original publications. If I regard it correctly, the original records are mostly deposited online and subsequent users should be able to cite the original datasets directly.

For datasets that have separate citations available, they have been added to the revision:

Bickert, T., Berger, W. H., Burke, S., Schmidt, H., and Wefer, G.: (Appendix A) Stable oxygen and carbon isotope ratios of Cibicidoides wuellerstorfi from ODP Hole 130-806B on the Ontong Java Plateau. PANGAEA [data set], https://doi.org/10.1594/PANGAEA.696408, 1993.

Chuang, C.; Lo, L., Zeeden, C., Chou, Y., Wei, K., Shen, C., Mii, H., Chang, Y., and Tung, Y.: Isotopic analysis of Globigerinoides sacculifer from ODP Hole 180-1115B, PANGAEA [data set], https://doi.org/10.1594/PANGAEA.899187, 2019.

Dang, H., Wu, J., Xiong, Z., Qiao, P., Li, T., and Jian, Z.: Oxygen isotopes of Globigerinoides ruber from core KX21-2 from the western equatorial Pacific over the last ~400 ka, PANGAEA [data set], https://doi.org/10.1594/PANGAEA.922658, 2020.

de Garidel-Thoron, T., Rosenthal, Y., Bassinot, F.C., and Beaufort, L.: Western Pacific Warm Pool Pleistocene paired d18O-Mg/Ca and SST reconstruction, NOAA National Centers for Environmental Information [data set], https://doi.org/10.25921/ejer-t729, 2005.

Holbourn, A. E., Kuhnt, W., Kawamura, H., Jian, Z., Grootes, P. M., Erlenkeuser, H., and Xu, J.: Stable isotopes on planktic foraminifera of sediment core MD01-2378, PANGAEA [data set], https://doi.org/10.1594/PANGAEA.263757, 2005.

Lo, L.: A dataset of the Mid-Brunhes period at site MD05-2925, Solomon Sea: Surface-subsurface planktonic foraminifera stable oxygen isotope and Mg/Ca ratios, Mendeley Data [data set], https://data.mendeley.com/datasets/9c2nnpchdh/1, 2021.

Lo, L., Chang, S., Wei, K., Lee, S., Ou, T., Chen, Y., Chuang, C., Mii, H., Burr, G. S., Chen, M., Tung, Y., Tsai, M., Hodell, D. A., and Shen, C.: Age model and oxygen isotopes of planktonic foraminifera from sediment core MD05-2925 off the Solomon Sea, PANGAEA [data set], https://doi.org/10.1594/PANGAEA.899216, 2019.

Medina-Elizalde, M., and Lea, D. W.: (Table S2) Stable oxygen isotope record and Mg/Ca ratios of Globigerinoides ruber from ODP Hole 130-806B. PANGAEA [data set], https://doi.org/10.1594/PANGAEA.772014, 2005.

Oppo, D.W., Linsley, B.K., Rosenthal, Y., Dannenmann, S., and Beaufort, L.: Sulu Sea core MD97-2141 foraminiferal oxygen isotope data, NOAA National Centers for Environmental Information [data set], https://doi.org/10.25921/qqqx-kt90, 2003.

Tian, J. Pak, Dorothy K., Wang, P., Lea, D. W., Cheng, X., and Zhao, Q.: (Appendix 1) Stable oxygen isotope ratios of Globigerinoides ruber and benthic foraminifera from ODP Site 184-1143, PANGAEA [data set], https://doi.org/10.1594/PANGAEA.707833, 2006.

Zhang, S., Yu, Z., Gong, X., Wang, Y., Chang, F., and Li, T.: Sable oxygen isotope and Mg/Ca ratios of planktonic foraminifera from KX97322-4 (KX22-4), PANGAEA [data set], https://doi.org/10.1594/PANGAEA.939377, 2021.

Comparison of the planktonic WPWP stack to the LR04 / LS16 benthic stacks: Why don't the authors compare planktonic and benthic δ18O records of the cores they are using to create the stack to get more direct assessments of the offsets between planktonic and benthic δ18O records? Benthic δ18O records are available for at least some of the cores.

The comparison of the new planktonic d18O WPWP stack to the timing of global average benthic d18O stacks provides justification for using LR04 as the initial alignment target for the WPWP stack. In the revised manuscript we will also add discussion of possible timing differences between WPWP planktonic and benthic d18O data based on 3 cores for which both proxies have been measured (MD05-2925, ODP 1143, and ODP 806). We will also add supplementary figures showing the planktonic and benthic d18O data from these cores plotted on their shared depth scales.

The revised manuscript reads (lines 244-248): "We also compare changes in planktonic and benthic $\delta^{18}$O measured within individual WPWP cores as a function of depth for MD05-2925, ODP-1143, and ODP-806 (Lo et al., 2019; Lo, 2021; Tian et al., 2006; Lea et al., 2000; Medina-Elizalde and Lea, 2005; Bickert et al., 1993). These cores do not show a consistent lead/lag between the planktonic $\delta^{18}$O and benthic $\delta^{18}$O records (**Fig. S1 - S3**), additionally indicating that the timing of WPWP planktonic and benthic $\delta^{18}$O change is similar on orbital time scales."

The authors present a stack record with a temporal resolution / time steps of 0.1 kyr, although the resolution of the individual records that go into the stack ranges between 0.33 and 2.3 kyr. The stack thus feigns a higher resolution than given, which should be avoided. The question is how this affects the statistical analyses presented in the article.

Our stack is produced with a temporal resolution of 0.1 kyr, but for spectral analysis the stack was first sub-sampled to have a 1 kyr resolution. This has been clarified in the methods section. Regarding other types of statistical analysis, the use of Gaussian process regression correctly reflects the relative uncertainty of stack values where data are relatively sparse compared to more densely sampled portions. This is why the stack's standard deviation is narrower in the more recent time interval (e.g., 0.15‰ for 0-60 ka) than for 500-800 ka, when fewer cores/measurements are available (as discussed on lines 234-238, now lines 275-278).

The revised manuscript reads (lines 186-187): "Both stacks were sub-sampled at 1 kyr spacing from 0 to 800 ka, and power spectral density was calculated using a multitaper power spectral density estimate function pmtm( ) in MATLAB,..."

The revised manuscript reads (lines 275-278): "...for 0 to 60 ka (Lee and Rand et al., 2023), while the new WPWP planktonic stack has a mean standard deviation of 0.16 ‰ for the same age range. A larger mean standard deviation of 0.19 ‰ for the full age range of 0 to 800 ka for our WPWP stack…"

The authors align their planktonic δ18O stack to the LR04 stack. They argue that there is almost no time shift between the planktonic and benthic records. Considering this, and by looking at the two stack records in comparison (Figure 4), the question arises, why scientists should use the planktonic δ18O stack instead of the LR04 or more recent regional LS16 benthic stacks in the future. I think that should be pointed out more clearly within the article.

We agree that the manuscript needs to more clearly describe the usefulness of the new WPWP stack as a regional alignment target that will improve relative age models and alignments of other planktonic WPWP records. We want this intended purpose to be easily identifiable for readers.

The similarity in timing of d18O change between the planktonic and benthic stacks provides justification for our use of the LR04 stack as an initial alignment target. However, there are differences in the amplitude of d18O change between the planktonic and benthic stacks and local WPWP signals that make the WPWP planktonic stack a preferable regional alignment target. We have added an example to the manuscript that demonstrates how the aligned age model for core MD01-2378 (from the Timor Sea and, thus, excluded from the new version of the stack) is improved when the WPWP planktonic stack is used for the alignment target compared to using the LR04 benthic stack for alignment. We also modified the manuscript's conclusion to clarify for which applications the WPWP planktonic stack is preferable to a benthic stack.

We have added a new Figure 7 and the following paragraph to Section 6.2.3 (lines 334-344):

"BIGMACS assumes that the records used for alignment share the same underlying signals; therefore, alignment should be more reliable with smaller uncertainties when a nearby planktonic $\delta^{18}$O record is aligned to the WPWP planktonic stack rather than the LR04 benthic stack. We demonstrate the potential impacts of aligning to different stacks by comparing the age estimates for the planktonic $\delta^{18}$O record of core MD01-2378 (Holbourn et al., 2005) from the Timor Sea (slightly outside the boundaries of the WPWP) based on alignment to either the WPWP stack or the LR04 stack (Bowman et al., 2023). Differences between the features of the two stacks during MIS 3 and 4 produce a ~14 kyr error in the alignment of the core to the LR04 stack, as indicated by the shifted position of the dashed vertical line in **Fig. 7**. The proper alignment of MIS 4 to the WPWP stack produces a 95% CI width of 6.5 kyr for estimated age at that time, compared to a 95% CI width of 13 to 18 kyr associated with the incorrect alignment to the LR04 stack. Because the planktonic $\delta^{18}$O records near the WPWP share features which differ from those of benthic $\delta^{18}$O, age model results for WPWP cores should be more accurate when their planktonic $\delta^{18}$O records are aligned to the WPWP stack than to a benthic stack."

The end of the conclusion section (lines 417-420) now reads: "Differences in glacial-interglacial amplitudes between the WPWP planktonic stack and benthic $\delta^{18}$O stacks validate that these differences are characteristic of planktonic $\delta^{18}$O throughout the WPWP. Furthermore, stratigraphic alignments of planktonic $\delta^{18}$O from cores near the WPWP

should produce more reliable relative age estimates when aligned to the WPWP planktonic stack instead of a benthic $\delta^{18}O$ stack."

If the δ18O records are shifted and scaled "to better match the target stack" (line 114), how useful is it to compare amplitudes or spectral power of the WPWP and LR04 stacks?

The phrasing used here was inadvertently confusing, and the revision explains this more clearly. (The BIGMACS "target stack" reflects planktonic values after the first iteration of alignments.) The mean and amplitude of the final WPWP stack match the mean and amplitude of the component planktonic records; therefore, comparison of the glacial cycle amplitudes and spectral power between the WPWP and LR04 are meaningful. The reported shift and scale factors indicate how much each individual record differs from the WPWP stack. (Thus, in Table 2 the shift values have an average of ~0 and the scale values average ~1.) This approach allows the amplitude of the stack to reflect variability in the common signal across sites although there are small constant offsets or amplification factors across the WPWP region.

The revised manuscript reads (lines 128-136): "In the second step, a stack is constructed with a Gaussian process regression over all $\delta^{18}O$ data using the aligned age models, and the stack's mean and amplitude are set to match the average values of the component records. The new stack is then used as the alignment target to construct age models for the next iteration, with alignment parameters updated to maximize likelihood using the Expectation Maximization (EM) algorithm. Iterations are performed until convergence. Core-specific shift and scale parameters (**Table 2**) reflect how much each individual record differs from the stack based on the assumption that all records share the same underlying signal but allowing for some scaling or offset based on consistent temperature/salinity gradients within the region as well as foraminiferal species differences (vital effects and depth habitat). Near-homogeneous planktonic $\delta^{18}O$ values between cores (and similar to the final stack) are indicated by shift parameters close to 0 and scale parameters close to 1."

Lines 142-144: "Because the stack alignment target is shifted and scaled to match its component records during each iteration, the final stack output by BIGMACS reflects the average WPWP planktonic $\delta^{18}O$ values, rather than the benthic $\delta^{18}O$ values of the initial target."

The authors introduce that they "seek to characterize WPWP climate on orbital timescales and its differences from high-latitude climate, which can help test hypotheses about the sensitivity of the WPWP to orbital forcing, ice volume, and greenhouse gas concentration" (lines 20-22). Later on, they compare their regional planktonic stack to a regional benthic stack, to the LR04 stack and to a regional WPWP SST stack record. They however miss to draw conclusions from their results. What can be inferred from the comparisons? What is, for instance, the value of comparing spectral power and variability of the regional planktonic stack to the global benthic stack?

We intentionally omitted any interpretation of WPWP climate mechanisms based on our understanding of the aims and scope of this journal, which provides guidance that "Any interpretation of data is outside the scope of regular articles" (https://www.earth-system-science-data.net/about/aims_and_scope.html). Further guidance on data description papers says: "Although examples of data outcomes may prove necessary to demonstrate data quality, extensive interpretations of data – i.e. detailed analysis as an author might report in a research article – remain outside the scope of this data journal. ESSD data descriptions should instead highlight and emphasize the quality,

usability, and accessibility of the dataset, database, or other data product and should describe extensive carefully prepared metadata and file structures at the data repository."
(https://www.earth-system-science-data.net/about/manuscript_types.html)

Therefore, we confine our discussion/conclusions to (1) the strengths and limitations of the methods used to create the stack (to provide guidance on how the data should be used for future studies), and (2) a brief description of how this stack differs from other available stacks that researchers might consider using.

In the revised version of the manuscript we have edited the conclusion to highlight the primary message of the publication. Specifically, the end of the last paragraph now read (lines 417-42): "Differences in glacial-interglacial amplitudes between the WPWP planktonic stack and benthic $\delta^{18}$O stacks validate that these differences are characteristic of planktonic $\delta^{18}$O throughout the WPWP. Furthermore, stratigraphic alignments of planktonic $\delta^{18}$O from cores near the WPWP should produce more reliable relative age estimates when aligned to the WPWP planktonic stack instead of a benthic $\delta^{18}$O stack."

This added conclusion is supported by a comparison of results for aligning planktonic d18O from core MD01-2378 to the WPWP planktonic stack versus the LR04 benthic stack (new Figure 7).

More specific comments:

Line 9: It should be "a smaller glacial-interglacial amplitude".

Updated, line 10.

Lines 19-20: "Thus, climate records of the WPWP region are expected to have features which differ from many other locations on Earth." This statement is rather general and should be more precise.

We agree. We revised this sentence to say (lines 21-23 "Thus, climate records of the WPWP region are expected to have features which differ from the high-latitude climate records often used to describe global climate change (e.g., Lisiecki & Raymo, 2005; Past Interglacials Working Group of PAGES, 2016)."

Added reference: Past Interglacials Working Group of PAGES (2016), Interglacials of the last 800,000 years, Rev. Geophys., 54, 162–219, doi:10.1002/2015RG000482.

Line 25: It might appear a bit old-fashioned, but don't foraminifers have tests instead of shells? Please correct.

Corrected, (now line 28).

Lines 24-35: Here, I clearly miss some references.

We agree with the reviewer that the previous introduction section did not have a sufficient number of references. The revised manuscript has more references, including more modern oceanography papers pertaining to the region. Lines 14-38:

[revised manuscript text omitted]

Line 78: There are calcification depth estimates from the study area (e.g. Hollstein et al., 2017). Why don't the authors consider these estimates? For instance, for G. ruber, the study indicates a calcification depth of 0-75 m, rather than the 30 m indicated by Wang et al. (2000).

Thank you for directing us to this additional publication. Additional information has been included within text.

The revised manuscript reads (lines 87-91): "All but one core in the stack uses $\delta^{18}O$ values measured from the planktonic species *Globigerinoides ruber (G. ruber)* sensu stricto (s.s.), whose depth habitat in the WPWP ranges from the upper 45 m to 105 m of the mixed layer depending on how calcification depth is calculated (Hollstein et al., 2017). One core, ODP 1115B, has data from a different planktonic species, *Trilobatus sacculifer* (formerly *Globigerinoides sacculifer*) whose depth habitat is 20 m to 75 m or potentially as deep as 45 m to 95 m (Sadekov et al., 2009; Hollstein et al., 2017)."

Line 79: Shouldn't it be *T. sacculifer*?

Updated. See response above.

Table 1: The reference of Chuang et al. (2018) is missing in the Reference list.

Corrected.

Line 119: The sentence is not complete.

Thank you for pointing out this error; "to have" should just be "have". The corrected sentence reads (lines 135-136): "Core sites with homogeneous planktonic $\delta^{18}O$ values have shift parameters close to 0 and scale parameters close to 1."

Lines 119-121: A short explanation for the shift and scale parameters might be helpful.

Descriptions of the record properties related to the shift and scale parameters (i.e., vertical offset and amplitude) have been added to the revised manuscript. Clarification that the shift and scale are relative to the planktonic WPWP stack has also been included. The new text reads (lines 142-144): "Because the stack alignment target is shifted and scaled to match its component records during each iteration, the final stack output by BIGMACS reflects the average WPWP planktonic $\delta^{18}O$ values, rather than the benthic $\delta^{18}O$ values of the initial target."

Lines 129-130: The authors do not apply local reservoir age corrections and assume a reservoir age standard deviation of 0.2 kyr. What is the rationale behind these assumptions?

The Marine20 calibration curve includes a time-dependent reservoir age which is used in radiocarbon data calibration (Heaton et al., 2020). We set the reservoir age offset (ΔR) to 0, meaning we did not change the reservoir age from the Marine20 default of 400 yrs. Additional description has been added to clarify that a reservoir age is still being used, as well as added notation for the reservoir age offset (ΔR). We assigned a 1-sigma uncertainty of 200 yr to the reservoir ages to account for possible changes to the reservoir age through time.

Our choices not to apply an offset from Marine20 and to use a 200-yr uncertainty are consistent with other WPWP studies. The originally published age model for MD05-2930 used the Marine09 calibration without a reservoir offset [Regoli et al., 2015]. Dang et al (2020) used the Marine13 calibration curve with offsets of <30 yr

and an uncertainty of <100 yr for core KX21-2. Importantly, the main goal of this manuscript is to provide a record of orbital-scale variability in planktonic d18O, it is not intended to provide 1-kyr precision of absolute ages and the revised manuscript clearly describes the limitations of the stack's age model. For example:

Lines 234-235: "The use of the LR04 stack as an initial alignment target for our WPWP stack assumes benthic and planktonic $\delta^{18}$O are changing synchronously"

Lines 250-257: "The relative timing of millennial-scale variability between the WPWP planktonic stack and benthic $\delta^{18}$O is more difficult to evaluate. … The portions of our WPWP stack older than 37 ka, which are not constrained by radiocarbon data, inherit the +/- 4 kyr age uncertainty of the LR04 stack used as the initial alignment target. Thus, we have no independent age estimates for WPWP planktonic $\delta^{18}$O change older than 37 ka."

Lines 135-139: If I understand it correctly, the authors choose a standard deviation of 1 kyr for the alignment of the records to MIS 3 and 4, but a standard deviation of 4 kyr for the first and last δ18O measurements of each core. I think, this needs some explanation.

We agree, and we have added more explanation in the revised manuscript. Specifically, the first and last additional ages with standard deviations of 4 kyr are age estimates taken from the core's previously published age models and given to BIGMACS to help aid age model construction. The prescribed uncertainty of 4 kyr accounts for potential age model differences in the core start/end ages between the BIGMACS age model and the previously published age models. In cores MD97-2141 and ODP 1115B, tie points were added with a smaller age uncertainty of 1 kyr to provide additional guidance to the BIGMACS alignment of specific features within the records to the BIGMACS stack. We are essentially correcting the default alignment by forcing certain features in the d18O record (MIS 3 and 4) to have certain ages, so we reduce the age standard deviation to 1 kyr.

These tie points in MIS 3 and 4 are necessary largely because our initial alignment target is the LR04 benthic d18O stack, which has different amplitudes for the isotopic stages than the planktonic d18O records being aligned. The revised manuscript will include an example of an alignment error at the MIS 4/5 transition that occurs when planktonic d18O from MD01-2378 is aligned to the LR04 stack; in contrast, the correct alignment is found when the WPWP planktonic stack is used for the alignment target.

Lines 166-171: "Core age models were also constrained by age estimates for the first and last $\delta^{18}$O measurement from each core based on previous publications. Because these previous age estimates were based on a variety of methods, they were assigned a Gaussian uncertainty with a relatively large standard deviation of 4 kyr. Additionally, we added tie points for two cores (ODP-1115B at 75 ka and MD97-2141 at 63 ka and 92.5 ka) to improve the alignment of Marine Isotope Stages (MIS) 3 and 4 to the target stack. Because these tie points were assigned based on identification of stratigraphic features in these two cores compared directly to the target stack, we assigned these age estimates a smaller standard deviation of 1 kyr."

Line 150: "Present day": Here the authors could be more precise, and indicate the year they are referring to.

The meaning of "present day" has been clarified in the revised manuscript to the time interval used for the World Oceans Atlas mean annual SST for the WPWP, which is 1955 - 2018  (Locarini et al., 2018).

The revised manuscript reads (lines 183 - 184): "Thus, the resulting $\Delta\delta^{18}$O$_{SST}$ measures change relative to mean annual SST of the WPWP from 1955 - 2018 (Locarini et al., 2018)."

Section 4.3: Please indicate, which software / tool was used to perform spectral analysis.

Updated in the manuscript methods and figure captions to indicate spectral analysis was performed using MATLAB's pmtm( ) function. The revised manuscript includes this information and a citation for Matlab.

Lines 185-188: "Power spectral density was calculated to quantify the strengths of response to orbital frequencies in $\delta^{18}$O for the WPWP and LR04 stacks. Both stacks were sub-sampled at 1 kyr spacing from 0 to 800 ka, and power spectral density was calculated using the multitaper power spectral density estimate function pmtm( ) in MATLAB, with the number of tapers set to two, a rate of one sample per kyr, and an nfft of 512 (The MathWorks Inc., R2023a)."

Line 185: "weaker the glacial-interglacial amplitude". Delete "the".

Corrected, the revised manuscript reads (lines 219-220): "The WPWP planktonic stack has weaker glacial-interglacial amplitudes than the global LR04 benthic $\delta^{18}$O stack (Lisiecki and Raymo, 2005) or a North Atlantic benthic $\delta^{18}$O stack produced by BIGMACS (Hobart et al., 2023)."

Line 254: "most negative". Please rephrase.

Updated in the revised manuscript which reads (line 295): "largest negative shifts"

Line 261: "where glacial surface water δ18Osw was more positive at the LGM". Please rephrase. I assume that the authors want to express that glacial δ18Osw was higher.

Updated in the revised manuscript which now reads (lines 302 - 303): "Unlike the central and southern WPWP where glacial surface water $\delta^{18}$O shifted toward more positive values at the LGM (Visser et al., 2003; Xu et al., 2008, Li et al. 2016)..."

Line 262: Which sites?

Thank you for pointing out the need to be more specific here, we replace "these cores" with specific cores.

The revised manuscript reads (lines 302 - 305): Unlike the central and southern WPWP where glacial surface water $\delta^{18}$O shifted toward more positive values at the LGM (Visser et al., 2003; Xu et al., 2008, Li et al. 2016), sites ODP-769A, MD97-2141, KX21-2, KX22-4 and ODP-806 show negative shifts in surface water $\delta^{18}$O at the LGM (Rosenthal et al., 2003; Lea et al., 2000).

Line 333: It should be "which has a higher resolution".

Corrected, line 409.

Tables 1 and 2: Could the authors maybe sort the entries of these tables?

This is an excellent suggestion. Cores have been sorted by longitude (west-to-east) in both tables and in Figure 2 for the revised manuscript.